

# Stochastic representation of the quantum quartic oscillator

**Gennaro Tucci[1★], Stefano De Nicola[2†], Sascha Wald[3,4‡] and Andrea Gambassi[5∘]**

**1** Max Planck Institute for Dynamics and Self-Organization, 37077 Göttingen, Germany
**2** ISTA, Am Campus 1, 3400 Klosterneuburg, Austria
**3** Statistical Physics Group, Centre for Fluid and Complex Systems,
Coventry University, Coventry, England
**4** $\mathbb{L}^4$ Collaboration & Doctoral College for the Statistical Physics of Complex Systems,
Leipzig-Lorraine-Lviv-Coventry, Europe
**5** SISSA - International School for Advanced Studies and INFN,
via Bonomea 265, I – 34136 Trieste, Italia

★ gennaro.tucci@ds.mpg.de , † stefano.de-nicola@ist.ac.at ,
‡ sascha.wald@coventry.ac.uk , ∘ gambassi@sissa.it

## Abstract

Recent experimental advances have inspired the development of theoretical tools to describe the non-equilibrium dynamics of quantum systems. Among them an exact representation of quantum spin systems in terms of classical stochastic processes has been proposed. Here we provide first steps towards the extension of this stochastic approach to bosonic systems by considering the one-dimensional quantum quartic oscillator. We show how to exactly parameterize the time evolution of this prototypical model via the dynamics of a set of classical variables. We interpret these variables as stochastic processes, which allows us to propose a novel way to numerically simulate the time evolution of the system. We benchmark our findings by considering analytically solvable limits and providing alternative derivations of known results.

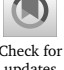

# 1 Introduction

Recent advances in cold atom experiments have motivated great theoretical interest in the non-equilibrium dynamics of isolated many-body quantum systems [1–10]. With the notable exception of integrable models [11], analytical insights into the time evolution of these systems are scarce. This motivates the search for novel numerical and analytical tools to analyze many-body quantum dynamics and to complement other widely used numerical approaches [12–14]. Some recent works [15–18] considered an exact representation of the dynamics of many-body quantum spin systems in terms of classical stochastic processes. This approach is based on a series of exact transformations, through which an interacting many-body quantum system can be exactly represented as non-interacting system under the action of a set of classical stochastic fields. In this stochastic approach, sometimes referred to as *disentanglement formalism*,[1] physical observables can be expressed as averages of classical functions over realizations of suitably constructed stochastic processes. This not only breaks down the complexity of many-body quantum interactions to that of classical stochastic differential equations but also allows one to bridge the gap between the quantum realm and classical stochastic processes, for which powerful numerical tools have been developed [20]. Beside this numerical application, the formalism can also be used as a field-theoretical tool to develop analytic expansions for observables [21]. The stochastic formalism has been mainly applied to many-body quantum spin systems [15–18, 21–23], where interactions are mapped to classical stochastic fields. Here, we explore a generalization of the stochastic approach to systems of interacting bosons. In particular we shall study the time-evolution of a zero-dimensional bosonic system with a non-linear interaction potential. These systems have attracted significant attention for their rich dynamical behavior and relevant experimental applications [24–29]. In particular, we shall consider the quantum quartic oscillator as a paradigmatic, non-linear bosonic quantum system [30–37].

---

[1]This terminology originates from the fact that the approach described here "disentangles" a time-ordered exponential, featured in the time-evolution operator, into a product of ordinary exponentials, see, e.g., Refs. [17, 19]. Hence, this term does not actually point to a relation to quantum entanglement. Here we mostly stick with the terminology "stochastic approach".

While any system with harmonic interactions can be mapped to non-interacting bosons, this is no longer the case for anharmonic potentials, whose simplest representative is arguably the quartic one.

In spite of its apparent simplicity, the quantum quartic oscillator is notoriously hard to study and involves subtle conceptual mathematical issues, even in its equilibrium formulation: notably, the divergence of the perturbation theory for the ground state energy in powers of the quartic coupling $\lambda$ and the fact that the energy levels have an infinite number of branch points in the complex $\lambda$-plane around the origin $\lambda = 0$, i.e., in the harmonic limit. From a physical perspective, these singularities are caused by level crossings in the eigenenergies [30, 31], leading to a non-trivial perturbation theory of the energy spectrum [36, 38]. The quantum quartic oscillator is also of practical relevance, as it can be used to approximate the low-energy behavior of more general, real-world quantum systems [39–41].

The quantum quartic oscillator has been investigated by approximation techniques, such as the semiclassical evaluation of its propagator [32–34, 37, 42] and, more recently, by looking at the evolution of relevant time-dependent observables [29, 43]. Interestingly, the exact form of the wave functions of the quartic oscillator has been reported only recently [44] but there is still no analytic expression for the corresponding quantized energy levels. Here, we show how the stochastic approach can be used to represent the dynamics of a quantum quartic oscillator in terms of ensembles of stochastically evolving harmonic oscillators. The representation we introduce is formally exact and can be used to develop analytical approximations or to evaluate numerically expectation values, providing a different viewpoint as well as a practical alternative to existing techniques.

This manuscript is organized as follows. In section 2 we derive a field-theoretical representation of the quantum quartic oscillator following earlier applications of the stochastic approach. This is done in two steps. First, the quartic part of the quantum quartic oscillator is decoupled by means of a Hubbard-Stratonovich transformation [45, 46], which casts the problem into that of the evolution of a quantum harmonic oscillator with time-dependent frequency. Second, a Lie-algebraic transformation is used to express the time-evolution operator in terms of ordinary exponentials featuring time-dependent classical coefficients [19, 47, 48], which we refer to as *stochastic variables*. The transformation yields a set of differential equations for the stochastic variables, which effectively describe the exact quantum dynamics of the quantum quartic oscillator. We also show that the formalism can be applied to quartic Hamiltonians with time-dependent coefficients. In section 3 we explain how expectation values of operators are calculated within this formalism, providing a general recipe and exact equations for a range of observables and initial states. The classical formulas we obtain are then benchmarked by considering two exactly solvable limits: the quantum harmonic oscillator, and the inherently classical limit in which all operators in the Hamiltonian commute with each other, corresponding to the disappearance of the kinetic energy in the Hamiltonian. In section 4, we elaborate on the stochastic interpretation of the present approach and discuss the conditions under which the field theory we obtain can be numerically simulated by means of stochastic processes. Building on this discussion, we show that for the quantum quartic oscillator with generic parameters (i.e., away from the two exactly solvable limits) our method can be benchmarked by applying a numerical stochastic scheme to evaluate time-evolved observables. In section 5 we adopt a field-theoretical viewpoint and develop a functional expansion of the time-evolution operator about the harmonic limit; we show that this is equivalent to the standard perturbative Dyson series. In section 6, we discuss how the semiclassical propagator and the partition function of the quantum quartic oscillator can be analytically recovered from the proposed field-theoretical picture. We present our conclusions in section 7, summarizing our results and outlining directions for further research. Several appendices cover mathematical details.

## 2  Stochastic Representation of the Quantum Quartic Oscillator

In this section, we introduce the stochastic approach briefly described in the introduction alongside with its novel application to bosonic systems. In particular, we focus on the quantum quartic oscillator, which is described by the Hamiltonian

$$\hat{H} \equiv \frac{\hat{p}^2}{2m} + \frac{1}{2}m\omega^2\hat{x}^2 + \frac{\lambda}{4}\hat{x}^4. \tag{1}$$

Here, the position operator $\hat{x}$ and the momentum operator $\hat{p}$ satisfy the canonical commutation relation $[\hat{x}, \hat{p}] = i\hbar$. The mass of the oscillator is denoted by $m$, the harmonic frequency by $\omega$ and the quartic potential is parametrized by the coupling constant $\lambda \geq 0$. Since an exact solution is not available for $\lambda \neq 0$, different approaches have been used in order to obtain insights on the quantum quartic oscillator, e.g., perturbation theory [30, 31, 36] and semiclassical approximations [32, 33, 42].

Here, we develop an alternative exact theoretical formulation of the problem, following the disentanglement approach recently applied to an ensemble of interacting quantum spin systems [15–18]. In particular, we investigate the unitary dynamics of the quantum quartic oscillator by exactly mapping it to stochastically driven operators, which have spin-like properties. This is done by employing a functional representation of the time-evolution operator

$$\hat{U}(t) \equiv \exp\left(-\frac{i}{\hbar}\hat{H}t\right). \tag{2}$$

First, we decouple the quartic term in the exponent of Eq. (2) via a Hubbard-Stratonovich transformation [45, 46] and trotterize [49, 50] the time-evolution operator on the time interval $\tau_n \equiv t/n$, in the limit $n \rightarrow \infty$, i.e.,

$$\hat{U}(t) = \lim_{n\to\infty}\left(\exp\left[-\frac{i\tau_n}{\hbar}\frac{\hat{p}^2}{2m}\right]\exp\left[-\frac{i\tau_n}{\hbar}\left(\frac{m}{2}\omega^2\hat{x}^2 + \frac{\lambda}{4}\hat{x}^4\right)\right]\right)^n. \tag{3}$$

To each of the $n$ Suzuki-Trotter factors appearing in Eq.(3), we apply a Hubbard-Stratonovich transformation which allows us to replace the quartic interaction with a quadratic one by introducing a real-valued auxiliary field $\phi$, i.e.,

$$\exp\left(-i\frac{\tau_n}{\hbar}\frac{\lambda}{4}\hat{x}^4\right) = \sqrt{\frac{\tau_n}{i\lambda\hbar\pi}}\int_{-\infty}^{\infty}d\phi\,\exp\left[\frac{i\tau_n}{\hbar}\left(\frac{\phi^2}{\lambda} - \hat{x}^2\phi\right)\right], \tag{4}$$

where we fix $\sqrt{i} = e^{i\pi/4}$ due to the positivity of $\lambda$, in order to ensure convergence. Equation (4) is derived in Appendix A.1. We can substitute the expression in Eq. (4) to each slice of the Suzuki-Trotter decomposition in Eq. (3), labeling the corresponding integration variable $\phi_k$ by $k \in \{1, \dots, n\}$. In the continuum limit $n \rightarrow \infty$, we can express $\hat{U}(t)$ as a functional integral with respect to the Hubbard-Stratonovich field $\phi(t)$. The corresponding measure is given by $\mathcal{D}\phi(t) \equiv \prod_k^n \sqrt{\tau_n/(i\lambda\hbar\pi)}\,d\phi_k$ in the limit $n \rightarrow \infty$, and we find

$$\hat{U}(t) = \int \mathcal{D}\phi\, e^{iS_0[\phi]}\,\mathbb{T}\exp\left[-\frac{i}{\hbar}\int_0^t d\tau\left(\frac{\hat{p}^2}{2m} + \frac{m}{2}\Omega^2(\tau)\hat{x}^2\right)\right], \tag{5}$$

where $\mathbb{T}\exp(\cdot)$ denotes the time-ordered exponential. The quadratic coupling is absorbed into an effective time-dependent real-valued frequency $\Omega^2(t)$, defined by

$$\Omega^2(t) \equiv \omega^2 + \frac{2\phi(t)}{m}, \tag{6}$$

and $S_0[\phi]$ denotes the Gaussian scalar action

$$S_0[\phi] = \frac{1}{\hbar\lambda} \int_0^t d\tau \, \phi^2(\tau). \tag{7}$$

Equation (5) casts the time-evolution operator of the quartic oscillator as an expectation value with respect to the Gaussian action $S_0[\phi]$ of the propagator of a harmonic oscillator with time-dependent frequency $\Omega(t)$. The associated effective Hamiltonian reads

$$\hat{H}_S(t) \equiv \frac{\hat{p}^2}{2m} + \frac{m}{2}\Omega^2(t)\hat{x}^2. \tag{8}$$

Despite the apparent simplification of dealing with the time-evolution operator of a quadratic Hamiltonian, time-ordering prevents the direct evaluation of the action of the operator in Eq. (5). This difficulty can be circumvented by means of a Lie-algebraic *disentanglement transformation* [15–17]. We define a new set of operators, whose linear combination with suitable coefficients reproduces the Hamiltonian in Eq. (1), i.e.,

$$\hat{S}^+ \equiv \frac{\hat{x}^2}{2\hbar}, \qquad \hat{S}^z \equiv i\frac{\{\hat{x}, \hat{p}\}}{4\hbar}, \quad \text{and} \quad \hat{S}^- \equiv \frac{\hat{p}^2}{2\hbar}. \tag{9}$$

These operators satisfy the commutation relations of the SU(2) algebra, viz.,

$$[\hat{S}^+, \hat{S}^-] = 2\hat{S}^z \quad \text{and} \quad [\hat{S}^z, \hat{S}^\pm] = \pm\hat{S}^\pm. \tag{10}$$

However, these operators differ from the conventional spin operators, since $\hat{S}^+ = (\hat{S}^+)^\dagger$ and $\hat{S}^- = (\hat{S}^-)^\dagger$ are Hermitian. The operators defined in Eq. (9) allow one to write the effective Hamiltonian as

$$\hat{H}_S(t) = \hbar\left[\frac{\hat{S}^-}{m} + m\Omega^2(t)\hat{S}^+\right]. \tag{11}$$

Following Refs. [19, 47, 48], we can express the time-ordered exponential in Eq. (5) as

$$\hat{U}_S \equiv \mathbb{T}\exp\left[-\frac{i}{\hbar}\int_0^t d\tau\left(\frac{\hat{p}^2}{2m} + \frac{1}{2}m\Omega^2(\tau)\hat{x}^2\right)\right] = e^{\xi^+(t)\hat{S}^+}e^{\xi^z(t)\hat{S}^z}e^{\xi^-(t)\hat{S}^-}, \tag{12}$$

with suitable "stochastic" variables $\xi^+, \xi^z$ and $\xi^-$. In section 4 we shall discuss the interpretation of these variables as stochastic processes, which motivates their naming.

The time evolution of $\xi^+$, $\xi^z$ and $\xi^-$ is obtained by imposing that the factorized expression on the right-hand side of Eq. (12) satisfies the same Heisenberg equation as $\hat{U}_S$ [19], leading to

$$i\frac{d\xi^+}{dt} + \frac{1}{m}(\xi^+)^2 = m\Omega^2(t), \qquad i\frac{d\xi^z}{dt} + \frac{2}{m}\xi^+ = 0, \qquad i\frac{d\xi^-}{dt} - \frac{e^{\xi^z}}{m} = 0. \tag{13}$$

The initial condition $\hat{U}(0) = \mathbb{1}$ imposes $\xi^+(0) = \xi^z(0) = \xi^-(0) = 0$. Note that, for a real-valued field $\phi$, $\xi^+$ and $\xi^-$ are purely imaginary while $\xi^z$ is real, implying that the exponential operators in Eq. (12) are unitary. We shall see that the stochastic variables are generally complex and thus the product of exponential operators in Eq. (12) is not unitary. The unitarity of the time-evolution operator is however recovered upon averaging, as described in Eq. (5). Substituting Eq. (12) into Eq. (5), allows one to express the time-evolution operator as

$$\hat{U}(t) = \left\langle e^{\xi^+(t)\hat{S}^+}e^{\xi^z(t)\hat{S}^z}e^{\xi^-(t)\hat{S}^-}\right\rangle_\phi, \tag{14}$$

where $\langle\ldots\rangle_\phi$ denotes the average with respect to the Gaussian field $\phi$, as defined in Eq. (5). The representation in Eq. (14) is exact and allows us to map the quantum dynamics onto

the time evolution of the stochastic parameters, see Eq. (13). Since this mapping is exact, the ensemble of trajectories $\xi^+(t)$, $\xi^z(t)$, $\xi^-(t)$ determined by the fields $\phi(t)$ encodes all the information about the underlying quantum problem. Moreover, Eq. (14) suggests that the time evolution is given by a weighted statistical average of successive actions of the exponential operators.

It is worth noting that the operators in Eq. (12) are the matrix elements of the covariance matrix

$$
\begin{pmatrix} \hat{x}^2 & \frac{1}{2}\{\hat{x}, \hat{p}\} \\ \frac{1}{2}\{\hat{x}, \hat{p}\} & \hat{p}^2 \end{pmatrix}, \tag{15}
$$

customarily used in the study of the dynamics of Gaussian wave packets under a quantum oscillator Hamiltonian [51, 52]. Indeed, as we shall further discuss below, the operators in Eq. (14) preserve the Gaussianity of a wave-packet and this justifies why it is convenient to work within this setting.

We can physically understand the action of the individual exponential operators in Eq. (12), and hence of $\hat{U}_S$, by studying their effect on a Gaussian wave packet $|\psi\rangle$, generally given by

$$
|\psi\rangle = \int_{-\infty}^{\infty} \frac{\mathrm{d}x}{(\pi\sigma^2)^{1/4}} \exp\left(-\frac{(x-a)^2}{2\sigma^2} + i(x-a)k\right) |x\rangle. \tag{16}
$$

The wave packet in Eq. (16) is parametrized by its average position $\langle x \rangle \equiv \langle \psi | \hat{x} | \psi \rangle = a$, variance $\langle x^2 \rangle_c \equiv \langle \psi | (\hat{x} - a)^2 | \psi \rangle = \sigma^2/2$ (where $\langle \dots \rangle_c$ denotes the connected component of the expectation value) and average momentum $\langle p \rangle \equiv \langle \psi | \hat{p} | \psi \rangle = k$. The variance of the momentum operator is given by $\langle p^2 \rangle_c \equiv \langle \psi | (\hat{p} - k)^2 | \psi \rangle = (2\sigma^2)^{-1}$. Here $|x\rangle$ denotes an eigenstate of the position operator and we set $\hbar = 1$. In order to visualize the action of the operators introduced in Eq. (12), it is convenient to see how the average and variance of the momentum and position operators transform. Namely, an operator of the type

- $\exp(\xi^- \hat{S}^-)$ acts on a Gaussian wave packet $|\psi\rangle$ by leaving the momentum cumulants $\langle p \rangle$ and $\langle p^2 \rangle_c$ unaltered, while shifting, respectively, the average and the variance of the position operator

$$
\langle x \rangle = a - \mathrm{Im}(\xi^-)k, \qquad \langle x^2 \rangle_c = \sigma^2/2 + (\mathrm{Im}(\xi^-)/\sigma)^2/2,
$$

  by terms that depend on the imaginary part of $\xi^-$;

- $\exp(\xi^z \hat{S}^z)$ rescales the $n$−th cumulant $\langle x^n \rangle_c$ of the position by a homogeneous constant $e^{-n\xi^z/2}$, while the $n$−th momentum cumulant $\langle \hat{p}^n \rangle_c$ by $e^{n\xi^z/2}$. For the first cumulants we have $\langle x \rangle = a\, e^{-\xi^z/2}$, $\langle x^2 \rangle_c = \sigma^2 e^{-\xi^z}/2$, $\langle p \rangle = k e^{\xi^z/2}$ and $\langle p^2 \rangle_c = e^{\xi^z}/(2\sigma^2)$. Thanks to the homogeneity of the transformation, the products $\langle x \rangle \langle p \rangle$ and $\langle x^2 \rangle_c \langle p^2 \rangle_c$ are preserved;

- $\exp(\xi^+ \hat{S}^+)$ leaves unaltered the position cumulants $\langle x \rangle$ and $\langle x^2 \rangle_c$, while shifting, respectively the average and the variance of the momentum operator

$$
\langle p \rangle = k + \mathrm{Im}(\xi^+)a, \qquad \langle p^2 \rangle_c = (2\sigma^2)^{-1} + (\mathrm{Im}(\xi^+)\sigma)^2/2,
$$

  by terms that depend on the imaginary part of $\xi^+$.

To further understand the physical significance of the action of the exponential operators in Eq. (12) on a Gaussian wave packet $|\psi\rangle$, it is useful to consider the exact phase space representation given by the Wigner function [53–55], defined as

$$
W(x, p) \equiv \frac{1}{\pi} \int \mathrm{d}y\, e^{-i2yp} \langle x + y | \psi \rangle \langle \psi | x - y \rangle. \tag{17}
$$

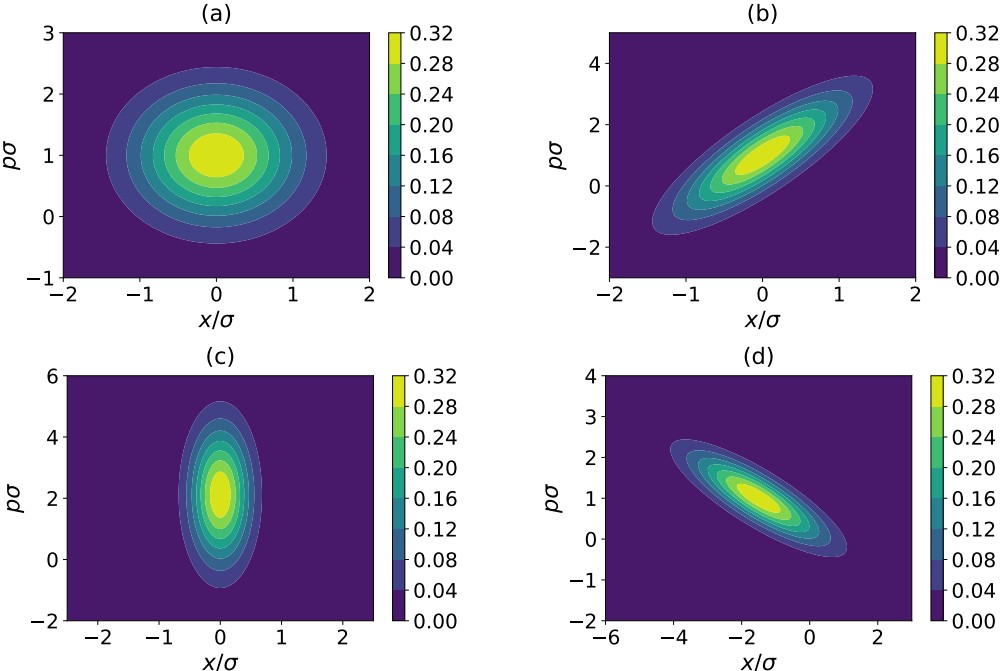

Figure 1: Contour plots of different Wigner functions. Panel (a) shows the Wigner function $W(x, p)$ for a Gaussian wave packet, see Eq. (18). In the $x$ and $p$ direction, $W(x, p)$ is a Gaussian centered around the phase space point $(a, k)$ with variance $\sigma^2/2$ and $(2\sigma^2)^{-1}$, respectively. We chose $a = 0$, $k = 1$, and $\sigma = 1$. Panel (b) shows the Wigner function $W_+(x, p)$ for the wave function $|\psi_+\rangle \equiv \exp(\xi^+ \hat{S}^+)|\psi\rangle$, see Eq. (82). This transformation shifts the $p$ variable by an $x$-dependent linear term, i.e., $p \to p - x \, \text{Im}(\xi^+)$; here we choose $\xi^+ = 1.5i$. Panel (c) shows the Wigner function $W_z(x, p)$ for the wave function $|\psi\rangle \equiv \exp(\xi^z \hat{S}^z)|\psi\rangle$, see Eq. (77). This transformation uniformly rescales the variables $(x, p)$ to $(x \, e^{\xi^z/2}, p \, e^{-\xi^z/2})$; here we chose $\xi^z = 1.5$. Panel (d) shows the Wigner function $W_-(x, p)$ for the wave function $|\psi\rangle \equiv \exp(\xi^- \hat{S}^-)|\psi\rangle$, see Eq. (71). This transformation shifts the $x$ variable by a $p$-dependent linear term according to $x \to x + p \, \text{Im}(\xi^-)$; we chose $\xi^- = 1.5i$.

The knowledge of $W(x, p)$ allows one to compute the expectation value of any operator $O(\hat{x}, \hat{p})$, expressed as a function of $\hat{x}$ and $\hat{p}$, as $\langle \psi | O(\hat{x}, \hat{p}) | \psi \rangle = \int \text{d}x \int \text{d}p \, W(x, p) O(x, p)$. This generalizes the results obtained above for the first cumulants of the wave packet, which are retrieved by identifying the operator $O(\hat{x}, \hat{p})$ with $\hat{x}$, $\hat{p}$, $\hat{x}^2$, $\hat{p}^2$, and their connected expectation value. Figure 1 shows how the Wigner function

$$W(x, p) = \frac{1}{\pi} \exp\left[ -\frac{(x - a)^2}{\sigma^2} - \sigma^2(p - k)^2 \right],$$ (18)

of a Gaussian wave packet is transformed upon the action of the three exponential operators in Eq. (12). In general, the exponential operators in Eq. (14) transform the parameters of the Gaussian wave packet by altering its original Heisenberg uncertainty relation $\langle \hat{x}^2 \rangle_c \langle \hat{p}^2 \rangle_c = 1/2$ [56], while they preserve the Gaussian structure of the relative Wigner function. For further details, see Appendix B.

# 3 Physical Observables

In this section, we illustrate how expectation values of observables can be expressed in the stochastic formalism by considering the position and momentum operator. In general, for a system prepared in a state $|\psi\rangle$, the expectation value of an observable $\hat{O}$ is given by $\langle\psi|\hat{O}|\psi\rangle$. In particular, we refer to $\langle\psi|\hat{O}^n|\psi\rangle$ as the $n$-th moment of the operator $\hat{O}$ with respect to the state $|\psi\rangle$. We can then express quantum expectation values as functional averages by replacing each time-evolution operator by its exact representation given by Eq. (14). This requires introducing independent Hubbard-Stratonovich fields $\phi$ and $\bar{\phi}$ for the two time-evolution operators. We refer to these fields as forward ($\phi$) and backward ($\bar{\phi}$) fields in analogy to the nomenclature of the Schwinger-Keldysh formalism [57–59].

## 3.1 Dynamics of a Gaussian Wave Packet

The dynamics of a particle in the presence of a quartic potential can be studied via the time-dependent moments of its position and momentum. We model the particle as a Gaussian wave packet, see Eq. (16), whose time evolution is governed by the time-evolution operator in Eq. (14). The evolved state is thus simply obtained by the subsequent action of the exponential operators in Eq. (14) and we find

$$|\psi(t)\rangle = \left\langle \int \mathrm{d}y \, \frac{\sqrt{\beta(t)\sigma}}{\pi^{1/4}} \exp\left( -\frac{\sigma^2 k^2}{2} - \frac{y^2}{2}\gamma(t) + i\sigma^2\beta(t)\mu y + \frac{\sigma^4\mu^2}{2}\alpha(t) \right) |y\rangle \right\rangle_\phi . \quad (19)$$

Here, we have introduced the generalized initial momentum $\mu$ of the wave packet as $\mu \equiv k - ia\sigma^{-2}$ and the variables $\alpha(t)$, $\beta(t)$ and $\gamma(t)$, which depend on the stochastic variables as

$$\alpha(t) \equiv \frac{1}{\sigma^2 - \xi^-(t)}, \qquad \beta(t) \equiv \frac{e^{\xi^z(t)/2}}{\sigma^2 - \xi^-(t)}, \qquad \gamma(t) \equiv \frac{e^{\xi^z(t)}}{\sigma^2 - \xi^-(t)} - \xi^+(t). \quad (20)$$

The time evolution of $\alpha$, $\beta$, and $\gamma$ is readily obtained from Eq. (13), i.e.,

$$i\frac{\mathrm{d}\gamma}{\mathrm{d}t} = \frac{\gamma^2}{m} - m\Omega^2, \qquad i\frac{\mathrm{d}\beta}{\mathrm{d}t} = \frac{\beta\gamma}{m}, \qquad i\frac{\mathrm{d}\alpha}{\mathrm{d}t} = \frac{\beta^2}{m}, \quad (21)$$

with initial conditions $\alpha(0) = \beta(0) = \gamma(0) = \sigma^{-2}$. These parameters are generally complex. Notably, the presence of $\sigma^2 > 0$ in Eq. (20) prevents divergences otherwise occurring for $\xi^-(t) = 0$, e.g., for $t = 0$.

The moments of the position operator on the Gaussian wave packet in Eq. (16) can be explicitly obtained by computing the expectation values $\langle\psi(t)|\hat{x}^n|\psi(t)\rangle$. To this end, we replace the time-evolved state $|\psi(t)\rangle$ by its representation involving the functional average $\langle\ldots\rangle_\phi$, derived in Eq. (14). A similar procedure is applied to $\hat{U}^\dagger(t)$, associated with the field $\bar{\phi}$ with action $-S_0[\bar{\phi}]$. We denote by $\alpha$, $\beta$ and $\gamma$ the solutions of Eqs. (21) that depend on the forward field $\phi$, and similarly we write $\bar{\alpha}$, $\bar{\beta}$, and $\bar{\gamma}$ for the solutions of the complex conjugates of Eqs. (21) associated with the backward field $\bar{\phi}$. Note that the fields $\phi$ and $\bar{\phi}$ are independent. The $n$-th moment of the position operator is finally found as

$$\langle\psi(t)|\hat{x}^n|\psi(t)\rangle = \left\langle \frac{2\sigma i^n \sqrt{\beta\bar{\beta}}}{(2\Gamma)^{(n+1)/2}} \mathrm{H}_n\left( \frac{\sigma^2\Delta}{\sqrt{2\Gamma}} \right) \exp\left[ -\sigma^2 k^2 + \frac{\sigma^4}{2}\left( A - \frac{\Delta^2}{\Gamma} \right) \right] \right\rangle_{\phi,\bar{\phi}}, \quad (22)$$

where we defined the auxiliary variables $\Gamma \equiv \gamma + \bar{\gamma}$, $\Delta \equiv \mu\beta - \mu^*\bar{\beta}$, $A \equiv \mu^2\alpha + (\mu^*)^2\bar{\alpha}$, and $\mathrm{H}_n$ denotes the $n$-th degree Hermite polynomial and $\mu^*$ the complex conjugate of $\mu$. Similarly,

the expression of the moments of the momentum operator are found to be

$$\langle \psi(t)| \hat{p}^n |\psi(t)\rangle = \left\langle 2\sigma \left[2\left(\frac{1}{\gamma}+\frac{1}{\bar{\gamma}}\right)\right]^{-(n+1)/2} i^n \sqrt{\frac{\beta\bar{\beta}}{\gamma\bar{\gamma}}} \, \mathrm{H}_n \left(-i\frac{\sigma^2\left(\mu\frac{\beta}{\gamma}+\mu^*\frac{\bar{\beta}}{\bar{\gamma}}\right)}{\sqrt{2\left(\frac{1}{\gamma}+\frac{1}{\bar{\gamma}}\right)}}\right) \right.$$

$$\left. \times \exp\left\{-\sigma^2 k^2 + \frac{\sigma^4}{2}\left[A-\mu^2\frac{\beta^2}{\gamma}-(\mu^*)^2\frac{\bar{\beta}^2}{\bar{\gamma}}+\frac{\left(\mu\frac{\beta}{\gamma}+\mu^*\frac{\bar{\beta}}{\bar{\gamma}}\right)^2}{\left(\frac{1}{\gamma}+\frac{1}{\bar{\gamma}}\right)}\right]\right\}\right\rangle_{\phi,\bar{\phi}}. \tag{23}$$

Note that these moments are formally retrieved from Eq. (22) by substituting

$$\gamma \to \gamma^{-1}, \quad \beta \to -i\beta/\gamma, \quad \text{and} \quad \alpha \to \alpha - \beta^2/\gamma.$$

As in the case of the time-evolution operator and the wave packet evolution, it is possible to express the dynamics of an observable as the expectation value of functions of the auxiliary parameters $\alpha$, $\beta$ and $\gamma$, which depend on the field $\phi$. Below, we will see how we may use these expressions for numerical calculations. As a final remark, we note that the convergence of Eq. (19) requires that $\mathrm{Re}(\gamma) > 0$, which follows naturally from the unitarity of the exponential operators in Eq. (12), see Appendix C for further details.

## 3.2 Exactly Solvable Cases

To the best of our knowledge, the non-linear evolution of the system of Eqs. (13) cannot in general be solved exactly. However, exact solutions can be found in two cases: the harmonic limit, in which Eqs. (13) become purely deterministic, and the *commuting limit*, in which the solutions of Eqs. (13) can be expressed in terms of the time integral of the field $\phi$. We use these exactly solvable instances as benchmarks for the stochastic formalism as well as for developing a physical intuition of its significance.

### 3.2.1 Harmonic case

In the case $\lambda = 0$ of the harmonic oscillator, the absence of the quartic term implies that Eqs. (13) reduce to the system of ordinary differential equations

$$i\frac{\mathrm{d}\xi^+}{\mathrm{d}t}+\frac{1}{m}(\xi^+)^2 = m\omega^2, \qquad i\frac{\mathrm{d}\xi^z}{\mathrm{d}t}+\frac{2}{m}\xi^+ = 0, \qquad i\frac{\mathrm{d}\xi^-}{\mathrm{d}t}-\frac{e^{\xi^z}}{m}=0, \tag{24}$$

which can be solved explicitly, i.e.,

$$\xi^+ = -im\omega\tan(\omega t), \qquad \xi^- = -\frac{i}{m\omega}\tan(\omega t), \qquad \xi^z = -\log\cos^2(\omega t). \tag{25}$$

This amounts to a known, exact parameterization of the quantum harmonic oscillator in terms of classical variables [19]. The time evolution of the stochastic variables $\xi^{+,-,z}$ for the harmonic oscillator shows periodic divergences at $t = t_n = \pi(1+2n)/(2\omega)$ with integer $n$, which, however, cancel out in the analytic computations of observables. This issue can be circumvented by equivalently considering the time evolution of the variables $\alpha$, $\beta$, $\gamma$, introduced in section 3.1. The solutions (25) can be plugged in the expressions of the observables, obtained in section 3, in order to compute exactly the corresponding dynamics. For instance, by inserting Eqs. (25) into Eq. (22), we retrieve the expressions of the moments of the position and

momentum operator for the Gaussian wave packet:

$$
\begin{aligned}
\langle \hat{x}^n \rangle &= \left( \frac{i}{2\sigma} \right)^n \left[ \sigma^4 \cos^2(\omega t) + x_0^4 \sin^2(\omega t) \right]^{\frac{n}{2}} \mathrm{H}_n \left( -\frac{i\sigma[a\cos(\omega t) + kx_0^2\sin(\omega t)]}{\sqrt{\sigma^4\cos^2(\omega t) + x_0^4\sin^2(\omega t)}} \right), \\
\langle \hat{p}^n \rangle &= \left( \frac{i}{2x_0^2\sigma} \right)^n \left[ x_0^4\cos^2(\omega t) + \sigma^4\sin^2(\omega t) \right]^{\frac{n}{2}} \mathrm{H}_n \left( -\frac{i\sigma[kx_0^2\cos(\omega t) - a\sin(\omega t)]}{\sqrt{x_0^4\cos^2(\omega t) + \sigma^4\sin^2(\omega t)}} \right),
\end{aligned}
\tag{26}
$$

where we introduced the typical harmonic oscillator length $x_0 \equiv (m\omega)^{-1/2}$.

### 3.2.2 Commuting limit

We study a particular case of the Hamiltonian in Eq. (1) in which the time evolution of the stochastic variables in Eq. (13) can be exactly solved. We consider the limit $m \to \infty$ with $m\omega^2$ held constant in order to ensure a constant finite energy. In this limit, the kinetic part of the Hamiltonian is suppressed and it coincides with the quartic potential. Accordingly, the Hamiltonian is only a function of the position operator and does not contain the momentum operator. In this sense we refer to this scenario as the *commuting limit*. Correspondingly, the non-linear terms in the differential equations (13) vanish, allowing us to express the explicit solution as

$$
\xi^+ = -i\omega^2 mt - 2i \int_0^t \mathrm{d}\tau\, \phi(\tau), \qquad \xi^- = \xi^z = 0,
\tag{27}
$$

where $\xi^+$ depends on the variable $\mathcal{W} \equiv \int_0^t \mathrm{d}s\, \phi(s)$, and allows us to express the average $\langle \cdots \rangle_\phi$ in Eq. (14) as an average with respect to the Gaussian weight of $\mathcal{W}$, i.e.,

$$
\hat{U}(t) = \int_{-\infty}^{+\infty} \mathrm{d}\mathcal{W} \frac{e^{i\mathcal{W}^2/(\lambda t)}}{\sqrt{i\pi\lambda t}} \exp\left[ -i\left( 2\mathcal{W} + m\omega^2 t \right) \hat{S}^+ \right].
\tag{28}
$$

From Eq. (28), it is apparent that, as expected in this case, the time evolution operator commutes with operators that depend only on the position operator $\hat{x}$. This implies the absence of dynamics for the particle position, compatibly with the vanishing kinetic energy. On the other hand, the moments of the momentum $\hat{p}$ grow in time, as a consequence of the Heisenberg uncertainty principle. We compute these moments by setting the auxiliary variables in Eq. (20) to $\alpha = \beta = \sigma^{-2}$ and $\gamma = \sigma^{-2} - i(2\mathcal{W} + \omega^2 mt)$ in Eq. (23) and by substituting the average $\langle \cdots \rangle$ with the integral $\int_{-\infty}^{+\infty} \mathrm{d}\mathcal{W}\, e^{i\mathcal{W}^2/(\lambda t)}/\sqrt{i\pi\lambda t}$ and similarly for the dual variables $\bar{\alpha}$, $\bar{\beta}$ and $\bar{\gamma}$. These expectation values can then be evaluated numerically, as we discuss in the next section.

## 4 Stochastic interpretation and numerical benchmark

In this section we show how the formalism presented above can be interpreted in terms of stochastic processes, which also allows us to benchmark our approach in cases where the model is not exactly solvable. In particular, by rotating the integration contour of the variable $\phi$ in the Hubbard-Stratonovich transformation in Eq. (4), after a generalization to the case of time-dependent couplings as in Eq. (68) of, cf., Appendix A.2, it is possible to show that

$$
\exp\left( -i\frac{\tau_n}{\hbar} \frac{\lambda_n}{4} \hat{x}^4 \right) = \sqrt{\frac{\tau_n}{\hbar\pi}} \int_{-\infty}^{\infty} \mathrm{d}\phi\, \exp\left[ -\frac{\tau_n}{\hbar} \left( \phi^2 + i\sqrt{i\lambda_n}\hat{x}^2\, \phi \right) \right],
\tag{29}
$$

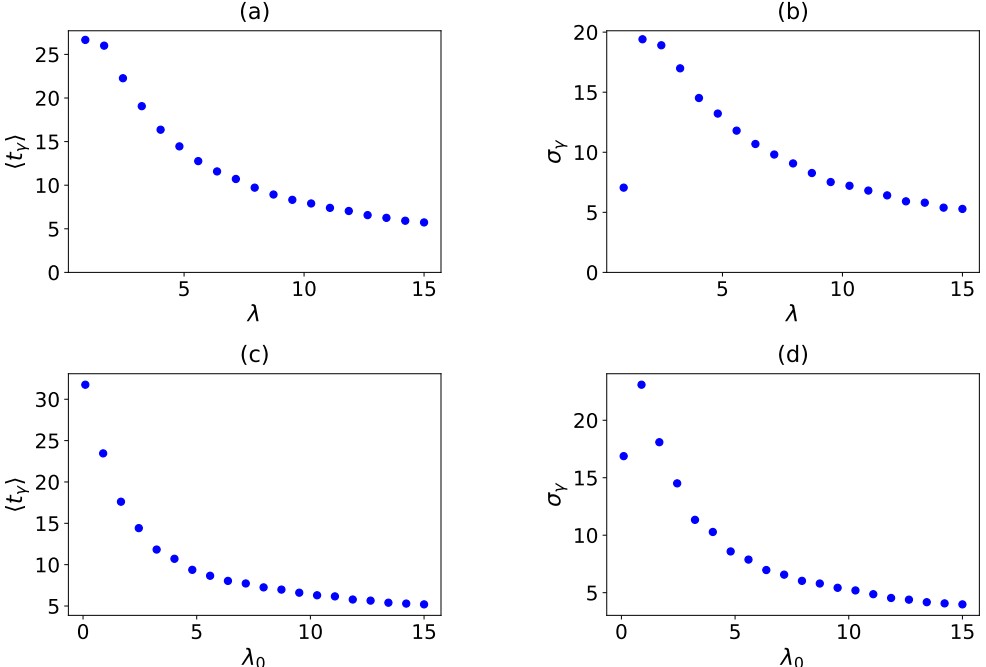

Figure 2: Panels (a) and (b) display a numerical estimate of the average value $\langle t_\gamma \rangle$ and the standard deviation $\sigma_\gamma \equiv \sqrt{\langle t_\gamma^2 \rangle - \langle t_\gamma \rangle^2}$ of the random variable $t_\gamma$, respectively, for the quartic oscillator, where $t_\gamma$ is the time at which $\mathrm{Re}(\gamma(t_\gamma)) = 0$ for the first time. We have used the Euler scheme with time step $\Delta t = 2/\pi \times 10^{-3}$ to simulate the time evolution of $\gamma$ with physical parameters are $\omega = 1$, $m = 10$, and extracted the value of $t_\gamma$ for $N = 10^4$ trajectories. Panels (c) and (d) show the same quantities and parameters except for the presence of a time-dependent potential $\omega^2(t) = \sin^2(t)$ and $\lambda(t) = \lambda_0 \sin^2(t)$. Note that, in general, $\langle t_\gamma \rangle \sim \sigma_\gamma$ at large values of $\lambda$: these strong fluctuations increase the probability of $\mathrm{Re}(\gamma(t_\gamma)) = 0$ at smaller values of $t_\gamma$ as the coupling $\lambda$ increases.

which follows from the change of variable $\phi / \sqrt{i} \to \phi$ in Eq. (68). Equation (29) allows one to represent $\hat{U}$ as

$$\hat{U}(t) = \int \mathcal{D}\phi \, e^{-S_0[\phi]} \, \hat{U}_S[\phi], \tag{30}$$

where $S_0$ is given by Eq. (7) and the time-evolution operator $\hat{U}_S$ in Eq. (12) displays the corresponding effective frequency $\Omega^2(t) \equiv \omega^2(t) + 2\sqrt{i\lambda(t)}\phi(t)/m$. The exponential of the action $S_0$ in Eq. (7) can be identified as a Gaussian probability measure for the field $\phi(t)$, whose time integral can be interpreted as a Wiener process [20]. Accordingly, Eqs. (13) can be understood as complex stochastic differential equations with a Gaussian white noise $\phi$. In this reformulation, however, the stochastic variables are generally complex, and do not preserve the unitarity of $\hat{U}_S$, which is only retrieved upon averaging. As a consequence, the expression inside the average $\langle \cdots \rangle_\phi$ of the observable expressions in Eq. (19) does not have to be convergent at all times and for all values of the quartic coupling. More precisely, since the effective frequency $\Omega^2(t)$ is complex, $\mathrm{Re}(\gamma)$ may attain a negative value after a certain time $t_\gamma > 0$, even though the initial value $\gamma(0) = \sigma^{-2} > 0$ is positive and real. One can numerically check that the average value $\langle t_\gamma \rangle$ of $t_\gamma$, interpreted as a random variable, decreases upon increasing the strength of the quartic coupling $\lambda$, see Fig. 2. Indeed, $t_\gamma$ is the first-passage time to the origin for the random variable $\mathrm{Re}(\gamma)$. This constitutes a limitation to the numerical application of the stochastic approach in the large-$\lambda$ regime, where $\langle t_\gamma \rangle$ is comparable with the standard

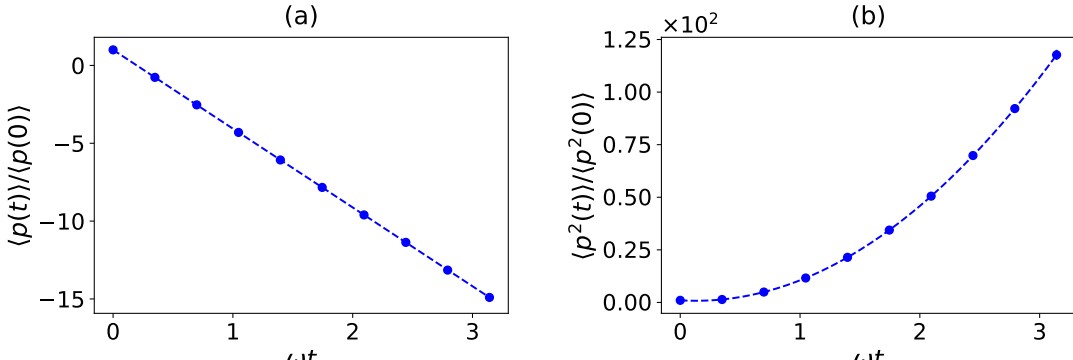

Figure 3: Time evolution of the first and second moment of the $\hat{p}$-operator for a Gaussian wavepacket with $\sigma = 0.5$, $k = 1.0$, $a = 0.5$ and evolving according to Eq. (1) with $\lambda = 0.2$ in the commuting limit. The dashed line is computed by integrating numerically the Schrödinger equation with the Crank-Nicholson method [60] with lattice spacing $\Delta x = 6 \times 10^{-4}$ and time step $\Delta t = \pi \times 10^{-4}$. Blue dots are computed with the stochastic method with $\Delta t = 10^{-4}$, and by sampling $N = 1.2 \times 10^{6}$ Gaussian random numbers. Error bars, corresponding to the statistical standard deviation over the sampling average, are not visible on the scale of the plot.

deviation $\sigma_{\gamma} \equiv \sqrt{\langle t_{\gamma}^2 \rangle - \langle t_{\gamma} \rangle^2}$. In this stochastic description, as reported in Appendix C, the divergences originate from the non-commutativity of the average $\langle \cdots \rangle_{\phi}$ over trajectories and the action of the operator $\hat{U}_S$ on a prescribed initial state $|\psi\rangle$ since, due to the non-unitarity of $\hat{U}_S$, the relation $\langle \hat{U}_S \rangle_{\phi} |\psi\rangle = \langle \hat{U}_S |\psi\rangle \rangle_{\phi}$ may not be satisfied. In the commuting limit, the above considerations translate in the simple change of variable $\mathcal{W}/\sqrt{i} \equiv \mathcal{W}'$ in Eq. (28), where this new $\mathcal{W}'$ can be interpreted a Gaussian random number with variance $\lambda t/2$. In spite of this limitation, the possibility to evaluate observables numerically by simulating classical stochastic dynamics allows us to further benchmark our approach. As a first check, we determine numerically the dynamics of the average momentum $\langle p \rangle$ for a Gaussian wave packet in the commuting limit. As we have shown, the stochastic differential equations are exactly solvable in this limit. It is thus possible to obtain directly the expressions for observables at a given time $t$ without having to integrate the time evolution numerically. These expressions are known functions of the time integral $\xi^+ = -2\sqrt{i}\mathcal{W}' - i\omega^2 mt$, which can be numerically simulated by drawing Gaussian random numbers with zero mean and variance given by $\lambda t/2$. Figure 3 shows the comparison of the numerical prediction of the dynamics of the first two moments of the position of a wave packet between the Crank-Nicholson method (dashed line) [60] used to integrate the Schrödinger equation numerically, and the prediction based on the stochastic interpretation of Eq. (30) (dots), finding good agreement. As a further validation of the presented stochastic description, we evaluate the dynamics of the expectation values in Eq. (22) within a range of parameters where no exact solutions are available. We determine our numerical results up to a time $t < \langle t_{\gamma} \rangle$, where no divergences are actually detected. For this purpose, we use an Euler discretization scheme [20] with time step $\Delta \tau = 10^{-5}$ to solve the complex-valued stochastic differential equations (13) for a given realization of the Wiener process $\phi(t)$. Once a sufficiently large number of realizations for the stochastic variables $\xi^+$, $\xi^z$, and $\xi^-$ or $\alpha$, $\beta$, and $\gamma$ are known, by averaging with respect to them, it is possible to compute the expectation value of a desired observable, see e.g., Eq. (22) or (23). Figure 4 shows the time evolution of the first moments of the position operator for a Gaussian wave packet for various choices of the parameters. In particular, we compare the numerical prediction of the stochastic method (dots) with standard integration of the Schrödinger equation with the

Cranck-Nicholson method (dashed line). Numerically, the proposed stochastic method has the advantage that the time-evolution of the many trajectories of the stochastic parameters, e.g., $\xi^{+,-,z}$, can be straightwardly parallelized. On the other hand, an increasing large number of realizations is needed in order to have accurate predictions for observables at longer times or larger quartic coupling strength $\lambda$, since fluctuations due to the noise grow correspondingly. This is similar to the behavior found for quantum spins systems [22].

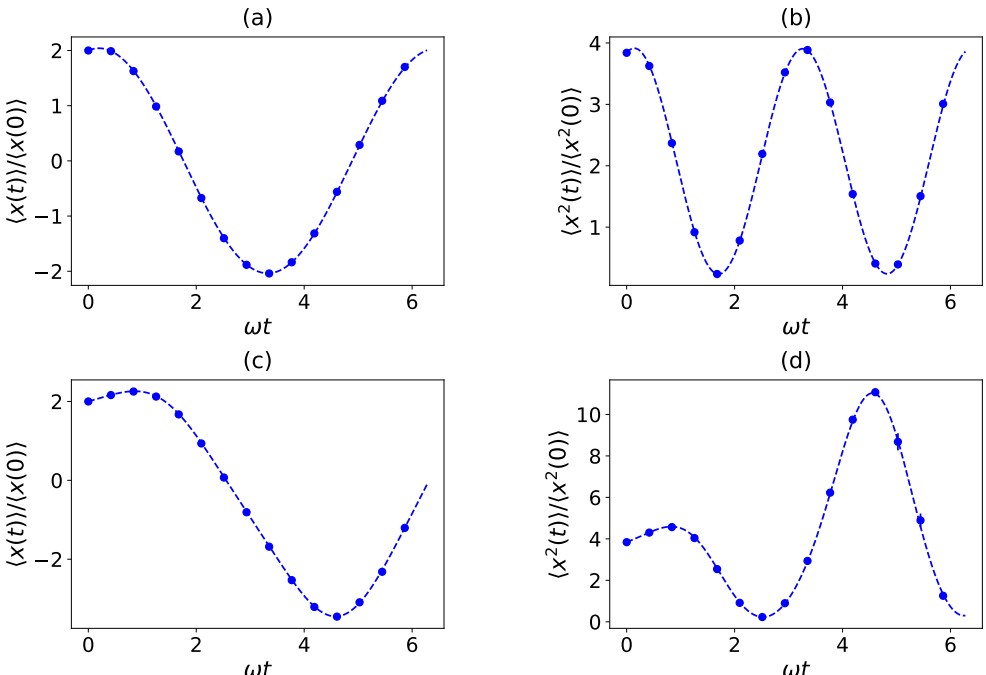

Figure 4: Time evolution of the expected (a) position $\langle x \rangle$ and (b) squared position $\langle x^2 \rangle$ of a Gaussian wave packet with $\sigma = 0.5$, $k = 1.0$,, and $a = 0.5$, evolving according to Eq. (1) with $m = 10$, $\omega = 1$, $\lambda = 0.2$. The dashed line corresponds to the numerics performed with the Crank-Nicholson method with space bin $\Delta x = 10^{-5}$ and time step $\Delta t = 10^{-3}$, while the blue dots are computed with the stochastic method with $\Delta t = 10^{-4}$, and sampling $N = 1.2 \times 10^5$ trajectories. The time $t$ has been chosen such that $t < \langle t_\gamma \rangle$ with $\langle t_\gamma \rangle \simeq 27$, and $\sigma_\gamma \simeq 19$, see Fig. 2a-b . Similarly, panels (c) and (d) display the evolution of $\langle x \rangle$ and $\langle x^2 \rangle$, respectively, in the case of $\omega^2(t) = \omega_0^2 \sin^2(t)$ and $\lambda(t) = \lambda_0 \sin^2(t)$ with $\omega_0 = 1$, $\lambda_0 = 0.2$, and $\langle t_\gamma \rangle \simeq 31$, with $\sigma_\gamma \simeq 18$, see Fig. 2c-d. As above, the dashed line corresponds to the numerics performed with the Crank-Nicholson method with space bin $\Delta x = 10^{-6}$ and time step $\Delta t = 10^{-3}$, while blue dots are computed with the stochastic method with $\Delta t = 10^{-4}$, and sampling $N = 1.2 \times 10^5$ trajectories. Error bars are given by statistical standard deviation over the trajectories average and are only visible in panel (d).

In summary, we have demonstrated that our formalism allows one to compute quantum observables for an interacting bosonic system from averaging classical stochastic processes, but we found that there are limitations to the applicability of this numerical technique. The mapping to stochastic processes also made it possible to further benchmark our approach in non-solvable cases. Since the stochastic description discussed in this section is formally identical to the field theory introduced in section 2, we refer to the present method as the "stochastic approach", although only its numerical application discussed here makes explicit use of stochastic processes.

## 5 Perturbative Expansion

In the stochastic approach, the time-evolution operator $\hat{U}$ in Eq. (14) is represented as a functional average over classical fields $\phi$. In this section, we show how this can be used to derive a perturbative expansion of $\hat{U}$ for the quartic oscillator in terms of the time-evolution operator $\hat{U}_0(t)$ of the harmonic case obtained for $\lambda = 0$. We start by rescaling the Hubbard-Stratonovich field as $\phi = \sqrt{\lambda}\,\varphi$ in the functional integral representation of $\hat{U}(t)$ in Eq. (5), yielding

$$\hat{U}(t) = \int \mathcal{D}\varphi \, e^{i\int_0^t ds\, \varphi^2(s)} \hat{U}_S\left[\sqrt{\lambda}\varphi\right]. \tag{31}$$

Here, the functional $\hat{U}_S[\phi]$ (which is also a function of time) is identified with the time-evolution operator of a harmonic oscillator with time-dependent frequency, given by Eq. (12). By Taylor-expanding the functional $\hat{U}_S$ around $\phi = 0$, corresponding to the harmonic time-evolution operator $\hat{U}_0$, and calculating the resulting Gaussian integrals, we get an asymptotic series for the propagator

$$\hat{U}(t) = \sum_{n=0}^{\infty}\left(i\frac{\lambda}{4}\right)^n \left(\prod_{m=1}^{n}\int_0^t dt_m\right) \frac{\delta^{2n}\hat{U}_S[\phi]}{\delta\phi(s_1)\cdots\delta\phi(s_{2n})}\bigg|_{\substack{\phi=0 \\ s_{2n-1}=s_{2n}=t_n}}, \tag{32}$$

where, on the right-hand side, only even orders of functional derivatives appear as a consequence of Wick's theorem, leaving the functional derivative of $\hat{U}_S$ evaluated at $\phi = 0$ as the only unknown. The series in Eq. (32) can be shown to be equivalent term by term to the Dyson series, see Appendix D. The equivalence of the functional expansion about $\phi = 0$ with the Dyson series allows us to use this functional formulation to calculate perturbative approximations of observables by field-theoretical means: we express the time evolution operators in the stochastic formalism, such that all operators are replaced by classical functionals, and then functionally expand about the non-interacting case. As it is usually the case in perturbative calculations, the asymptotic series in Eq. (32) is expected to fail whenever we consider states for which the quartic term $\lambda\,\hat{x}^4/4$ is not negligible relative to the harmonic Hamiltonian. Indeed, it is a well-known fact that the Dyson series of the quartic oscillator has a vanishing radius of convergence [30, 31].

## 6 Semiclassical Approximation

In this section, we show how the semiclassical approximation for the propagator associated with $\hat{U}(t)$ in the representation of Eq. (14) and the partition function of the system in Eq. (1) can be expressed within the present formalism. Other than giving us an additional benchmark for the theory, this shows that it is possible to find an alternative description of the stationary trajectories contributing to the semiclassical approximation for the quartic oscillator.

### 6.1 Propagator

The propagator $G(x_f, t|x_i, 0)$, which gives the probability amplitude for a particle located at $x_i$ at the initial time $t_i = 0$ to reach the position $x_f$ at time $t_f = t$, is defined by

$$G(x_f, t|x_i, 0) \equiv \langle x_f|\hat{U}(t)|x_i\rangle. \tag{33}$$

The semiclassical approximation of the propagator for the quantum quartic oscillator has been extensively studied in the literature, see, e.g., Ref. [32] for an overview. Here, we show how

to express $G(x_f, t|x_i, 0)$ in terms of the stochastic variables. This expression can be derived by inserting the representation of $\hat{U}$ in Eq. (14), by acting on an eigenstate of the position $|x_i\rangle$ according to Eq. (86) in Appendix C, and finally by projecting on $\langle x_f|$. This leads to

$$G(x_f, t|x_i, 0) = \left\langle \frac{1}{\sqrt{-2\pi\hbar\xi^-}} \exp\left[\frac{\xi^z}{4} + \frac{\xi^+ x_f^2}{2\hbar} + \frac{(x_f e^{\xi^z/2} - x_i)^2}{2\hbar\xi^-}\right] \right\rangle_\phi, \quad (34)$$

where $\xi^{+,-,z}$ are evaluated at time $t$. Note that the stochastic variables in the above expression are functions of $\phi$ evaluated at the final time $t = t_f - t_i$, while the initial and final position $x_i$ and $x_f$ are fixed parameters.

In the harmonic case $\lambda = 0$, by explicit substitution of Eq. (25) into Eq. (34), we retrieve the known expression of the harmonic propagator $G_{\mathrm{HO}}$, i.e.,

$$G_{\mathrm{HO}}(x_f, t|x_i, 0) = \sqrt{\frac{m\omega}{2i\pi\hbar\sin(\omega t)}} \exp\left[\frac{im\omega}{2\hbar\sin(\omega t)}\left((x_f^2 + x_i^2)\cos(\omega t) - 2x_i x_f\right)\right]. \quad (35)$$

In the quartic case, inside the average in Eq. (34) we recognize a different way of representing the propagator of an harmonic oscillator with time-dependent frequency $\Omega^2(t)$. It is well-known that in the case of quadratic interactions, even in the time-dependent case, the propagator can be expressed in a closed form through the contributions arising from classical paths, see, e.g., Ref. [32]. Accordingly, the propagator of a harmonic oscillator with generic time-dependent frequency can be reformulated as

$$G_{\mathrm{HO}}(x_f, t|x_i, 0) = G_{\mathrm{HO}}(0, t|0, 0) \exp\left[\frac{i}{\hbar} S_{\mathrm{HO}}(x_f, t|x_i, 0)\right]. \quad (36)$$

Here, the classical action $S_{\mathrm{HO}}$ of the harmonic oscillator is given by

$$S_{\mathrm{HO}}(x_f, t|x_i, 0) \equiv \frac{m}{2}\int_0^t d\tau \left[\dot{x}^2(\tau) - \Omega^2(\tau)x^2(\tau)\right], \quad (37)$$

and is computed along the classical path $x(\tau)$ which satisfies the Euler-Lagrange equation

$$\ddot{x}(\tau) + \Omega^2(\tau)x(\tau) = 0, \quad (38)$$

with boundary conditions $x(0) = x_i$ and $x(t) = x_f$. The prefactor $G_{\mathrm{HO}}(0, t|0, 0)$ is given by

$$G_{\mathrm{HO}}(0, t|0, 0) = \sqrt{\frac{m}{2\pi i\hbar f(t)}}, \quad (39)$$

where the *density of paths* $f(t)$ is obtained, according to Gelfand-Yaglom formula [61], as a solution of the differential equation in Eq. (38) with $x \to f$, with initial conditions $f(0) = 0$ and $\dot{f}(0) = 1$. Note that the propagator $G_{\mathrm{HO}}(0, t|0, 0)$ can be represented via the Feynman path integral associated to the quadratic action in Eq. (37) with boundary conditions $x_f = x_i = 0$, which entails that the function $f(t)$ is proportional to the determinant of the linear operator $d_t^2 + \Omega^2(t)$ expressed through Eq. (39) [32].

The solutions of Eq. (38) depend on the realization of the field $\phi$ which enters $\Omega$ according to Eq. (6). Alternatively, they can be expressed in terms of the stochastic variables according to

$$x(\tau|t) = \frac{e^{-\xi^z(\tau)/2}}{\xi^-(t)}\left\{x_f \xi^-(\tau)e^{\xi^z(t)/2} + x_i\left[\xi^-(t) - \xi^-(\tau)\right]\right\},$$
$$f(\tau) = im\,\xi^-(\tau)e^{-\xi^z(\tau)/2}, \quad (40)$$

where we emphasize that $t$, $x_i$, and $x_f$ are fixed parameters, and $\tau \in [0, t]$ is a variable. In turn, $\xi^{+,-,z}$ depend on $\phi$ via Eq. (13). In order to simplify the notation, the dependence of $x$ and $f$ on $x_i$ and $x_f$ and the functional dependence on $\phi$ are omitted.

Finally, by inserting Eq. (36) into (34), the propagator $G(x_f, t|x_i, 0)$ for the quartic oscillator reads

$$
\begin{aligned}
G(x_f, t|x_i, 0) &= \int \mathcal{D}\phi \ \sqrt{\frac{m}{2\pi i \hbar f(t)}} \exp\left[ \frac{i}{\hbar\lambda} \int_0^t \phi^2(\tau)d\tau + \frac{i}{\hbar} S_{\text{HO}}(x_f, t|x_i, 0) \right] \\
&= \int \mathcal{D}\phi \ \exp\left[ \frac{i}{\hbar\lambda} \int_0^t \phi^2(\tau)d\tau \right] G_{\text{HO}}[\{\xi(t)\}].
\end{aligned}
\tag{41}
$$

As anticipated, Eq. (41) illustrates the fact that the propagator for the quartic oscillator is given by an infinite collection of classical path contributions of harmonic oscillators with different time dependent frequencies. Moreover, Eq. (41) provides the starting point to perform the semiclassical approximation, corresponding to the limit $\hbar\lambda \to 0$. We start by rescaling spatial coordinates as $y \equiv x\sqrt{\lambda}$, and we set $y_f \equiv \sqrt{\lambda}x_f$, $y_i \equiv \sqrt{\lambda}x_i$. Due to the homogeneity of $S_{\text{HO}}$ with respect to $x_i$ and $x_f$, this rescaling allows us to cast the propagator $G$ as

$$
G(x_f, t|x_i, 0) = \int \mathcal{D}\phi \ \sqrt{\frac{m}{2\pi i \hbar f(t)}} \exp\left[ \frac{i}{\hbar\lambda}\left( \int_0^t \phi^2(\tau)d\tau + S_{\text{HO}}(y_f, t|y_i, 0) \right) \right].
\tag{42}
$$

In the limit $\hbar\lambda \to 0$, we can approximate the functional integral by applying the stationary phase method [62]. We obtain

$$
\bar{\phi}(\tau) = -\frac{1}{2} \frac{\delta S_{\text{HO}}(y_f, t; y_i, 0)}{\delta\phi(\tau)}\bigg|_{\bar{\phi}} = \frac{\bar{y}^2(\tau)}{2} = \lambda\frac{\bar{x}^2(\tau)}{2}.
\tag{43}
$$

By inserting Eq. (43) in Eq. (38) we retrieve the equation for the classical trajectories of the quartic oscillator

$$
\ddot{\bar{x}}(\tau) + \omega^2\bar{x}(\tau) + \frac{\bar{x}^3(\tau)}{m} = 0,
\tag{44}
$$

with boundary conditions $\bar{x}(0) = x_i$ and $\bar{x}(t) = x_f$. The solution of Eq. (44) can be expressed in terms of Jacobi elliptic functions [42]. Note that, according to Eqs. (43) and (44), the stationary field $\bar{\phi}$ is continuous and twice differentiable, meaning that among all possible realizations of $\phi$ only a subset with sufficient regularity contributes in the semiclassical limit. As reported in Ref. [33], the associated stationary trajectories $\bar{x}$ can be classified in terms of the sign of the momentum $\hat{p}$ of the particles $\bar{p} = m\dot{\bar{x}}$ at the boundary points $x_i$ and $x_f$.

The semiclassical approximation is obtained by considering terms of the expansion in $\sqrt{\hbar\lambda}$ up to the second order around the stationary phase solution. In this spirit, we introduce the change of variable $\phi = \bar{\phi} + \sqrt{\hbar\lambda}\,\varphi$ and truncate the expansion around $\bar{\phi}$ at second order in $\varphi$, leading to

$$
G(x_f, t|x_i, 0) = \sum_k \sqrt{\frac{m}{2\pi i \hbar f_k(t)}} \ e^{\frac{i}{\hbar}S_k(x_f, t|x_i, 0)} \int \mathcal{D}\varphi \exp\left[ i \iint_0^t dt_1 dt_2 \, \varphi(t_1) H_k(t_1, t_2)\varphi(t_2) \right],
\tag{45}
$$

with $\mathcal{D}\varphi \equiv \prod_n d\phi_n/\sqrt{i\pi}$, where the second functional derivative of the action computed at $\bar{\phi}$ corresponds to the operator

$$
H_k(t_1, t_2) \equiv \delta(t_1 - t_2) + \frac{1}{2} \frac{\delta^2 S_{\text{HO}}(y_f, t|y_i, 0)}{\delta\phi(t_1)\delta\phi(t_2)}\bigg|_{\bar{\phi}_k};
\tag{46}
$$

the zeroth order of the expansion renders the classical action $S_k(x_f, t|x_i, 0)$ of the quartic oscillator

$$S_k(x_f, t | x_i, 0) = \int_0^t \mathrm{d}\tau \left[ \frac{m}{2} (\dot{x}_k)^2 - \frac{1}{2} m \omega^2 x_k^2 - \frac{\lambda}{4} x_k^4 \right], \tag{47}$$

evaluated on $x_k$, the $k$-th solution of Eq. (44) with $\phi_k \equiv \bar{\phi}[x_k]$, and $f_k \equiv f[\phi_k]$. In order to determine the functional Gaussian integral in Eq. (45) it is necessary to calculate the determinant of the operator $H_k$. As shown in Appendix E, this computation can be done explicitly, leading to the final expression of the semiclassical propagator as

$$G(x_f, t | x_i, 0) = \sum_k \sqrt{\frac{m}{2\pi i \hbar F_k(t)}} \, e^{iS_k(x_f, t | x_i, 0)/\hbar}, \tag{48}$$

where $F_k \equiv F[\phi_k]$, similarly to $f_k$, satisfies the differential equation

$$\ddot{F}_k(\tau) + \left[ \omega^2 + 3 \frac{\lambda}{m} x_k^2(\tau) \right] F_k(\tau) = 0, \tag{49}$$

with $F_k(0) = 0$ and $\dot{F}_k(0) = 1$, being $F_k$ proportional to the determinant of the $H_k$ operator.

In the process of deriving Eq. (47) within the present approach, we relate the operator $H_k(t_1, t_2)$ to the functional derivative of (38) according to Eq. (110) in Appendix E, leading to $\det(H_k) = \sqrt{f_k(t)/F_k(t)}$. In summary, we have derived an alternative representation of the propagator $G(x_f, t | x_i, 0)$ of the quartic oscillator, expressed as a weighted collection of the propagators of effective harmonic oscillators $G_{\mathrm{HO}}(x_f, t | x_i, 0)$. Moreover, we provided a parametrization of the time-evolution of these harmonic oscillators in terms of the stochastic variables. Finally, we have proven that the semiclassical approximation of $G(x_f, t | x_i, 0)$ relies on the calculation of the determinant of the second variation of the effective action $S_{\mathrm{HO}}$, and how this determinant is linked to the *density of paths* along the classical trajectory $\bar{x}_k$ of the quartic oscillator.

## 6.2 Partition Function

In this section we show that our formulation is not only restricted to non-equilibrium problems, but it can be used to extract finite-temperature [15, 16] or ground state [21] properties by Wick-rotating to imaginary time. Here we provide an additional confirmation of the validity of the stochastic representation of the quantum quartic oscillator by obtaining the semiclassical limit ($\hbar \to 0$) of its partition function $Z(\beta)$, where $\beta$ is the inverse temperature. The partition function $Z(\beta)$ is obtained from the propagator $G(x_f, t | x_i, 0)$ in Eq. (34) according to [62]

$$Z(\beta) = \int_{-\infty}^{+\infty} \mathrm{d}x \, G(x, t = -i\hbar\beta | x, 0) = \int_{-\infty}^{+\infty} \mathrm{d}x \left\langle \frac{1}{\sqrt{-2\pi\hbar\xi^-}} \exp\left[ \frac{\xi^z}{4} + \frac{x^2}{2\hbar} \left( \xi^+ + \frac{(e^{\xi^z/2} - 1)^2}{\xi^-} \right) \right] \right\rangle_\phi. \tag{50}$$

The associated evolution of the stochastic variables as functions of $\beta$ is determined by the set of differential equations

$$\frac{1}{\hbar} \frac{\mathrm{d}\xi^+}{\mathrm{d}\beta} - \frac{(\xi^+)^2}{m} + m\omega^2 + 2\phi = 0, \qquad \frac{1}{\hbar} \frac{\mathrm{d}\xi^z}{\mathrm{d}\beta} - \frac{2}{m} \xi^+ = 0, \qquad \frac{1}{\hbar} \frac{\mathrm{d}\xi^-}{\mathrm{d}\beta} + \frac{e^{\xi^z}}{m} = 0, \tag{51}$$

with the usual initial conditions $\xi^{+,-,z}(0) = 0$. Equations (51) suggest that we may retrieve the semiclassical limit by retaining the leading-order contributions of the series expansion of $\xi^{+,-,z}$ in integer powers of $\hbar$, i.e.,

$$\xi^+(\beta) = \sum_{k=1}^\infty \hbar^k f_k(\beta), \qquad \xi^z(\beta) = \sum_{k=1}^\infty \hbar^k g_k(\beta), \qquad \xi^-(\beta) = \sum_{k=1}^\infty \hbar^k l_k(\beta). \tag{52}$$

By inserting these expansions in Eq. (51) and by retaining terms up to order $O(\hbar^3)$, we get closed-form expressions for the first coefficients. We find that $f_2 = g_1 = g_3 = l_2 = 0$ vanish and the non-vanishing contributions read

$$
\begin{aligned}
f_1 &= -m\omega^2\beta - 2\int_0^\beta \mathrm{d}\tau\,\phi(\tau), \\
f_3 &= \frac{1}{m}\int_0^\beta \mathrm{d}\tau(f_1(\tau))^2, \\
g_2 &= \frac{2}{m}\int_0^\beta \mathrm{d}\tau f_1(\tau),
\end{aligned}
\qquad
\begin{aligned}
l_1 &= -\frac{\beta}{m}, \\
l_3 &= -\frac{2}{m^2}\int_0^\beta \mathrm{d}\tau \int_0^\tau \mathrm{d}s f_1(s).
\end{aligned}
\tag{53}
$$

It follows that all the coefficients in Eq. (53) are expressed in terms of $f_1(\beta)$, so that the calculation reduces to the evaluation of moments of $f_1$. We now consider the leading contribution up to order $O(\hbar^2)$ of Eq. (50) by explicitly inserting the expansions in Eq. (52):

$$
Z(\beta) = \int_{-\infty}^{+\infty}\mathrm{d}x\left\langle\frac{1}{\hbar}\sqrt{\frac{m}{2\pi\beta}}\exp\left(f_1\frac{x^2}{2}\right)\left\{1+\frac{\hbar^2}{2}\left[\frac{g_2}{2}-\frac{l_3}{l_1}+\frac{x^2}{2}\left(f_3+\frac{g_2^2}{4l_1}\right)\right]+o(\hbar^2)\right\}\right\rangle_\phi. \tag{54}
$$

The expectation value with respect to the Gaussian field $\phi$ is then easily calculated and it is given by the sum of the following expressions

$$
\begin{aligned}
\left\langle\exp\left(f_1\frac{x^2}{2}\right)\frac{g_2}{2}\right\rangle_\phi &= -e^{-\beta V(x)}\frac{\beta^2(m\omega^2+\lambda x^2)}{2m}, \\
\left\langle\exp\left(f_1\frac{x^2}{2}\right)\frac{l_3}{l_1}\right\rangle_\phi &= -e^{-\beta V(x)}\frac{\beta^2(m\omega^2+\lambda x^2)}{3m}, \\
\left\langle\exp\left(f_1\frac{x^2}{2}\right)f_3\right\rangle_\phi &= \beta^2\frac{e^{-\beta V(x)}}{m}\left[\frac{\beta(m\omega^2+\lambda x^2)^2}{3}-\lambda\right], \\
\left\langle\exp\left(f_1\frac{x^2}{2}\right)\frac{g_2^2}{4l_1}\right\rangle_\phi &= \beta^2\frac{e^{-\beta V(x)}}{m}\left[-\frac{\beta(m\omega^2+\lambda x^2)^2}{4}+\frac{2}{3}\lambda\right],
\end{aligned}
\tag{55}
$$

where $V(x) \equiv \frac{1}{2}m\omega^2 x^2 + \frac{\lambda}{4}x^4$. Given that the average is computed with respect to a quadratic measure with zero average, the odd moments vanish and the contributions in Eq. (55) are real-valued. By collecting all of the above terms, we finally obtain the semiclassical expansion of the partition function $Z(\beta)$:

$$
\begin{aligned}
Z(\beta) &= \int_{-\infty}^{+\infty}\mathrm{d}x\,e^{-\beta V(x)}\left\{1+\frac{\hbar^2\beta^2}{12m}\left[x^2\beta\frac{(m\omega^2+\lambda x^2)^2}{2}-(m\omega^2+3\lambda x^2)\right]+o(\hbar^2)\right\} \\
&= \int_{-\infty}^{+\infty}\mathrm{d}x\,e^{-\beta V(x)}\left\{1+\frac{\hbar^2\beta^2}{12m}\left[\beta\frac{(V'(x))^2}{2}-V''(x)\right]+o(\hbar^2)\right\},
\end{aligned}
\tag{56}
$$

which matches the expression reported in the literature, see, e.g., Ref. [62]. As in the case of the propagator $G(x_f, t|x_i, 0)$ discussed in the previous section, we have shown how it is possible to represent the partition function $Z(\beta)$ of the quartic oscillator in terms of the imaginary-time version of the stochastic variables.

## 7 Summary and outlook

In this work we generalized the stochastic formalism recently introduced for quantum spin systems [15–18] to the case of non-linear bosonic systems, explicitly considering the quantum

quartic oscillator. We derived the exact disentangled representation of the time-evolution operator of the quartic oscillator in Eq. (14) and provided exact formulas for the time evolution of Gaussian wave packets. In particular, we considered the time evolution of the expectation values of the position and of the momentum operator and their corresponding higher moments. We benchmarked our approach (i) in the harmonic and the commuting limit by comparison with the respective analytic solutions and (ii) for a quartic anharmonicity the comparison was done numerically by using the stochastic interpretation of the formalism. We further use the stochastic formalism to derive a perturbative expansion of the time-evolution operator in powers of the quartic term. We recover the usual Dyson series for the quantum quartic oscillator which thus implies that our formalism is viable for evaluating perturbative expansions of observables. Finally, we provided a semiclassical expansion of the propagator and the partition function. Our results agree with known expressions, proving the validity of this alternative formulation.

The stochastic approach presented in this work provides a novel theoretical formulation of the quantum quartic oscillator as a paradigm of non-linear bosonic systems. We described the quartic oscillator by an ensemble of harmonic oscillators under the influence of classical (stochastic) fields. This exact representation is a suitable starting point for developing a range of approximations, which we illustrated in sections 5 and 6. Furthermore, the realizations of the classical stochastic fields fully encode the underlying quantum problem. Hence, their study should provide information about the dynamical properties of the corresponding quantum system, as recently found for spin systems [18]. Finally, the stochastic approach establishes a connection between bosonic quantum systems and the theory of classical stochastic processes. In particular, this connection allows us to evaluate the evolution of physical observables numerically in a novel fashion, e.g., our method does not require a truncation of the Hilbert space dimension. However, despite this numerical strategy being quite intuitive and simple to implement, the non-unitarity of the effective time-evolution operator leads to artificial divergences such that simulations break down after a finite time which depends on the strength of the quartic coupling.

Possible further directions include the generalization of our approach to coupled oscillators [63] and bosonic lattice systems. This could be done by decoupling interactions between different sites by means of additional Hubbard-Stratonovich fields, as is done for quantum spin systems [15–18]. The disentanglement approach could then provide a numerical technique to simulate bosonic dynamics as well as an analytical framework based on which further approximations can be developed.

## Acknowledgments

**Funding information** S. De Nicola acknowledges funding from the Institute of Science and Technology Austria (ISTA), and from the European Union's Horizon 2020 research and innovation program under the Marie Skłodowska-Curie Grant Agreement No. 754411. S. De Nicola also acknowledges funding from the EPSRC Center for Doctoral Training in Cross-Disciplinary Approaches to NonEquilibrium Systems (CANES) under Grant EP/L015854/1.

# A Stochastic integral and time-dependent quartic Hamiltonian

## A.1 Gaussian integral

In this section, we derive Eq. (4), which is fundamental for the construction of the stochastic description of the quartic potential. First, we evaluate the integral

$$\oint_C dz\, e^{iaz^2}\,, \tag{57}$$

where $a > 0$ and the contour $C$ is displayed in Fig. 5, see, e.g., [64]. Because of the absence

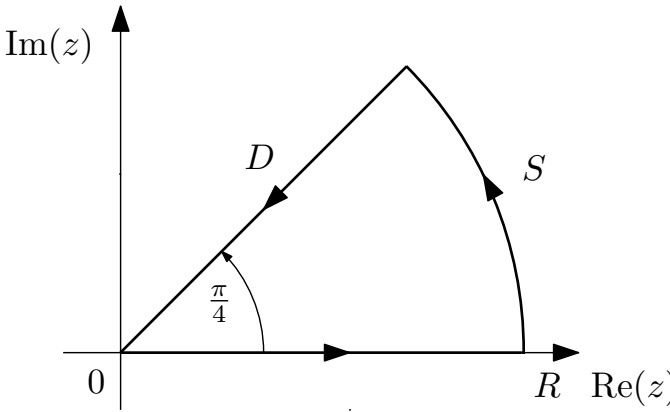

Figure 5: Integration contour $C$ of Eq. (57). $C$ consists of a circular section of angle $\pi/4$ and radius $R$, including the arc $S$ and the segment $D$, in addition to the segment $[0, R]$ on the real line.

of singularities inside the contour $C$, the residue theorem immediately implies

$$\oint_C dz\, e^{iaz^2} = 0\,, \tag{58}$$

which results from the sum of the following contributions:

$$\oint_C dz\, e^{iaz^2} = \int_0^R dx\, e^{iax^2} + \int_S dz\, e^{iaz^2} + \int_D dz\, e^{iaz^2} = 0\,, \tag{59}$$

where $S$ represents the circular arc of radius $R$, parametrized as $z = R\,e^{i\phi}$ with $\phi \in (0, \pi/4)$, and $D$ the radial contribution corresponding to $z = r\,e^{i\pi/4}$ with $r \in (0, R)$, in the direction shown in Fig. 5. Finally, we are interested in the limit $R \to \infty$. The integral along $D$ can be expressed as

$$\int_D dz\, e^{iaz^2} = e^{i\pi/4} \int_R^0 dr\, \exp\left\{ia\left(r\,e^{i\pi/4}\right)^2\right\} = -e^{i\pi/4} \int_0^R dr\, e^{-ar^2}\,, \tag{60}$$

coming from the change of variables $z = r\,e^{i\pi/4}$. Similarly, the integral along $S$ is given by

$$\int_S dz\, e^{iaz^2} = iR \int_0^{\pi/4} d\phi\, \exp\left\{iaR^2\left[\cos(2\phi) + i\sin(2\phi)\right]\right\}\,, \tag{61}$$

and it can be shown to vanish in the limit $R \to \infty$. In fact, we start from the inequality

$$\left| \int_S dz \, e^{iaz^2} \right| \le R \int_0^{\pi/4} d\phi \, e^{-aR^2 \sin(2\phi)} = \frac{R}{2} \int_0^{\pi/2} d\theta \, e^{-aR^2 \sin\theta}, \tag{62}$$

where last equality follows from introducing $\theta \equiv 2\phi$. Moreover, for $\theta \in (0, \pi/2)$, we have that $2\theta/\pi \le \sin\theta \le \theta$, yielding

$$\left| \int_S dz \, e^{iaz^2} \right| \le \frac{R}{2} \int_0^{\pi/2} d\theta \, e^{-aR^2 \sin\theta} \le \frac{R}{2} \int_0^{\pi/2} d\theta \, e^{-2aR^2\theta/\pi} = \frac{\pi}{4aR} \left( 1 - e^{-aR^2} \right), \tag{63}$$

which vanishes for $R \to \infty$. Finally, we get

$$\int_0^\infty dx \, e^{iax^2} = e^{i\pi/4} \int_0^\infty dr \, e^{-ar^2} = e^{i\pi/4} \sqrt{\frac{\pi}{4a}} = \sqrt{\frac{i\pi}{4a}}, \tag{64}$$

where we have fixed $\sqrt{i} = e^{i\pi/4}$. This last result can be generalized to

$$\int_{-\infty}^{+\infty} dx \, e^{i(ax^2+bx)} = \sqrt{\frac{i\pi}{a}} e^{-ib^2/(4a)}. \tag{65}$$

## A.2 Time-dependent quartic Hamiltonian

In the framework discussed in section 2, it is natural to generalise the expression of Eq. (14) to the case of the time-dependent Hamiltonian

$$\hat{H}(t) = \frac{\hat{p}^2}{2m} + \frac{1}{2} m\omega^2(t)\hat{x}^2 + \frac{\lambda(t)}{4} \hat{x}^4, \tag{66}$$

with a non-negative quartic coupling, $\lambda(t) \ge 0$. The procedure follows the same steps as in the time-independent case, i.e.,

(i) we perform a Trotter-Suzuki splitting of the time-evolution operator, yielding

$$\hat{U}(t) = \lim_{n\to\infty} \left\{ \exp\left[ -\frac{i\tau_n}{\hbar} \frac{\hat{p}^2}{2m} \right] \exp\left[ -\frac{i\tau_n}{\hbar} \left( \frac{m}{2}\omega_n^2 \hat{x}^2 + \frac{\lambda_n}{4}\hat{x}^4 \right) \right] \right\}^n, \tag{67}$$

with $\omega_n^2 \equiv \omega^2(n\tau_n)$ and $\lambda_n \equiv \lambda(n\tau_n)$;

(ii) this is followed by the Hubbard-Stratonovich transformation, performed through an integral of the type

$$\exp\left( -i\frac{\tau_n}{\hbar} \frac{\lambda_n}{4} \hat{x}^4 \right) = \sqrt{\frac{\tau_n}{i\hbar\pi}} \int_{-\infty}^\infty d\phi \, \exp\left[ \frac{i\tau_n}{\hbar} \left( \phi^2 - \sqrt{\lambda_n} \hat{x}^2 \phi \right) \right]; \tag{68}$$

(iii) Finally, combining all previous steps, we retrieve Eq. (14) with $\Omega^2(t) \equiv \omega^2(t) + 2\sqrt{\lambda(t)}\phi(t)/m$ and $S_0[\phi] = \hbar^{-1} \int_0^t d\tau \, \phi^2(\tau)$.

Once again, this allows one to study the dynamics of a quantum problem by studying a set of classical differential equations.

# B Action of $\exp(w\hat{S}^{\pm,z})$ on a Gaussian wave packet

We now investigate the action of the operators appearing in Eq. (12), i.e.,

$$\hat{U}^\alpha \equiv \exp(w\hat{S}^\alpha)\,, \tag{69}$$

with complex $w$ and $\alpha \in \{+,-,z\}$, on the Gaussian wave packet $|\psi\rangle$, reported in Eq. (16). For a general complex-valued $w$, the exponential operators are not unitary and do not conserve the normalization of the state. For simplicity, we consider here values of $w$ for which the corresponding $\hat{U}^\alpha$ conserves the state normalization. Note that the $\hat{S}^\alpha$ operators in Eqs. (9) are, at most, of quadratic order with respect to the operators $\hat{x}$ and $\hat{p}$. In what follows, we assume $\hbar = 1$ and a real-valued $w$. In order to analyze the action of $\hat{U}^\alpha$ in Eq. (69), we have introduced in Eq. (17) the Wigner function $W(x,p)$ for the generic state $|\psi\rangle$.

$W(x,p)$ provides a phase space description of the state, and allows us to compute expectation values of operators of the type $O(\hat{x},\hat{p})$ as $\int \mathrm{d}x \int \mathrm{d}p\, O(x,p)W(x,p)$ [53–55]. The evaluation of the Wigner function in Eq. (18) for the Gaussian wave packet $|\psi\rangle$ in Eq. (16) is obtained by substitution of Eq. (16) into Eq. (17) and integrating with respect to $y$. First, we consider the action of the operator $\hat{U}^+ = \exp(iw\hat{S}^+)$ on $|\psi\rangle$, that we denote as $|\psi_+\rangle \equiv U^+|\psi\rangle$. The state $|\psi_+\rangle$ is simply given by

$$|\psi_+\rangle = \int \frac{\mathrm{d}x}{\sqrt[4]{\pi\sigma^2}} \exp\left[-\frac{(x-a)^2}{2\sigma^2} + iw\frac{x^2}{2} + ik(x-a)\right]|x\rangle\,, \tag{70}$$

which follows from the fact that $\hat{S}^+$ acts trivially on its eigenstate $|x\rangle$.

By substituting Eq. (70) into (17) we get the Wigner function for $|\psi_+\rangle$

$$W_+(x,p) = \frac{1}{\pi} \exp\left[-\frac{(x-a)^2}{\sigma^2} - \sigma^2(p-k-wx)^2\right]\,, \tag{71}$$

which is equal to $W$ up to position-dependent shift in the momentum. Accordingly, the expectation value of operators of the form $O(\hat{x})$ is unaffected by the $\hat{U}^+$ transformation, i.e., $\langle\psi|O(\hat{x})|\psi\rangle = \langle\psi_+|O(\hat{x})|\psi_+\rangle$. On the other hand, the expectation value of a momentum-dependent operator $O(\hat{p})$ can be expressed as

$$\begin{aligned}
\langle\psi_+|O(\hat{p})|\psi_+\rangle &= \int \mathrm{d}x \int \mathrm{d}p\, O(p)W_+(x,p) \\
&= \sqrt{\frac{\sigma^2}{\pi(1+\sigma^4 w^2)}} \int \mathrm{d}p\, O(p) \exp\left[-\sigma^2\frac{(p-k-wa)^2}{1+\sigma^4 w^2}\right] \\
&= \sqrt{\frac{\sigma^2}{\pi}} \int \mathrm{d}q\, O\left(\sqrt{1+\sigma^4 w^2}(q-k)+wa+k\right)e^{-\sigma^2(q-k)^2} \\
&= \langle\psi|O\left(\sqrt{1+\sigma^4 w^2}(\hat{p}-k)+wa+k\right)|\psi\rangle\,,
\end{aligned} \tag{72}$$

where the second line comes from direct substitution of Eq. (71) and the final result from the change of variable $p - wa - k = (q-k)\sqrt{1+\sigma^4 w^2}$. Equation (72) tells us that expectation values with respect to the state $|\psi_+\rangle$ of operators depending only on $p$ are equivalent to expectation values with respect to the Gaussian wave packet $|\psi\rangle$ with the rescaled and shifted momentum operator $\sqrt{1+\sigma^4 w^2}(\hat{p}-k)+wa+k$. In particular, for the mean and the variance of the momentum operator we can immediately read off from Eq. (72) that

$$\begin{aligned}
\langle p\rangle &\equiv \langle\psi_+|\hat{p}|\psi_+\rangle = k+wa\,, \\
\langle p^2\rangle_c &\equiv \langle\psi_+|\hat{p}^2|\psi_+\rangle - \langle\psi_+|\hat{p}|\psi_+\rangle^2 = \frac{1+\sigma^4 w^2}{2\sigma^2}\,.
\end{aligned} \tag{73}$$

These parameters, together with the unaltered cumulants of the position operator, allow us to fully characterize the state $|\psi_+\rangle$. As an explicit time-dependent example, we consider the evolution of the wave packet under the action of the harmonic oscillator Hamiltonian. Referring to Eqs. (25), we find that

$$
\begin{aligned}
w(t) &= -m\omega \tan(\omega t), \\
\langle p(t) \rangle &= k - am\omega \tan(\omega t), \\
\langle p^2(t) \rangle_c &= \frac{1 + (m\omega\sigma^2)^2 \tan^2(\omega t)}{2\sigma^2}.
\end{aligned}
\tag{74}
$$

Next, we consider the case of the operator $\hat{U}^z = \exp(iw\{\hat{x}, \hat{p}\}/4)$ whose action on the Gaussian wave packet $|\psi\rangle$, which we denote by $|\psi_z\rangle \equiv \hat{U}^z |\psi\rangle$, reads

$$
|\psi_z\rangle = \int \frac{\mathrm{d}x}{\sqrt[4]{\pi\sigma^2}} \exp\left[ \frac{w}{4} - \frac{(x\,e^{w/2} - a)^2}{2\sigma^2} + ik(x\,e^{w/2} - a) \right] |x\rangle .
\tag{75}
$$

This is computed considering the direct action of $\hat{S}^z$ on $|x\rangle$ according the property of the dilation operator $e^{by\frac{d}{dy}} f(y) = f(e^b y)$, where $f$ is any sufficiently smooth function, similarly to what has been done for Eq. (86). The Wigner function $W_z(x, p)$ of the state $|\psi_z\rangle$ can be directly evaluated as

$$
W_z(x, p) = \frac{1}{\pi} \exp\left[ -\frac{(x\,e^{w/2} - a)^2}{\sigma^2} - \sigma^2(p\,e^{-w/2} - k)^2 \right].
\tag{76}
$$

Equation (75) shows that the action of $\hat{U}^z$ consists in a uniform rescaling all the $x$ variables by a factor $e^{w/2}$.

As for $|\psi_z\rangle$, the Wigner function $W_z$ is equivalent to $W$ up to a rescaling of the variables. It follows that the expectation value of an operator $O(\hat{x})$, depending only on $x$, is given by

$$
\begin{aligned}
\langle \psi_z | O(\hat{x}) | \psi_z \rangle &= \int \mathrm{d}x \int \mathrm{d}p\, O(x) W_z(x, p) \\
&= \frac{e^{w/2}}{\sqrt{\pi\sigma^2}} \int \mathrm{d}x\, O(x) \exp\left[ -\frac{(x\,e^{w/2} - a)^2}{\sigma^2} \right] \\
&= \langle \psi | O(\hat{x}\,e^{-w/2}) | \psi \rangle ,
\end{aligned}
\tag{77}
$$

and, analogously, for a $p$−dependent operator $O(\hat{p})$, we get

$$
\langle \psi_z | O(\hat{p}) | \psi_z \rangle = \langle \psi | O(\hat{p}\,e^{w/2}) | \psi \rangle ,
\tag{78}
$$

which reflects the rescaling action of $\hat{U}^z$. It follows that the first connected moments of $\hat{x}$ and $\hat{p}$ on $|\psi_z\rangle$ are given by

$$
\begin{aligned}
\langle x \rangle &\equiv \langle \psi_z | \hat{x} | \psi_z \rangle = a\,e^{-w/2}, \\
\langle x^2 \rangle_c &\equiv \langle \psi_z | \hat{x}^2 | \psi_z \rangle - \langle \psi_z | \hat{x} | \psi_z \rangle^2 = \frac{\sigma^2}{2} e^{-w}, \\
\langle p \rangle &\equiv \langle \psi_z | \hat{p} | \psi_z \rangle = k\,e^{w/2}, \\
\langle p^2 \rangle_c &\equiv \langle \psi_z | \hat{p}^2 | \psi_z \rangle - \langle \psi_z | \hat{p} | \psi_z \rangle^2 = \frac{e^w}{2\sigma^2}.
\end{aligned}
\tag{79}
$$

These quantities fully characterize the state $|\psi_z\rangle$. According to Eqs. (25), for a $|\psi\rangle$ evolving under the effect of an harmonic oscillator Hamiltonian, we have

$$w(t) = -\log\cos^2(\omega t),$$

$$\langle x(t)\rangle = a\cos(\omega t), \quad \langle p\rangle(t) = \frac{k}{\cos(\omega t)},$$

$$\langle x^2(t)\rangle_c = \frac{\sigma^2}{2}\cos^2(\omega t), \quad \langle p^2(t)\rangle_c = \frac{1}{2\sigma^2\cos^2(\omega t)}. \tag{80}$$

Finally, we consider the action of the operator $\hat{U}^- = \exp\left(iw\hat{p}^2/2\right)$ on the wave packet $|\psi\rangle$; the resulting state $|\psi_-\rangle \equiv \hat{U}^-|\psi\rangle$ is found to be

$$|\psi_-\rangle = \sqrt{\frac{\sigma^2}{\sigma^2 - iw}} \int \frac{dx}{(\pi\sigma^2)^{1/4}} \exp\left[-\frac{(x - a - ik\sigma^2)^2}{2(\sigma^2 - iw)} - \frac{\sigma^2 k^2}{2}\right]|x\rangle. \tag{81}$$

The associated Wigner function $W_-(x, p)$ reads

$$W_-(x, p) = \frac{1}{\pi}\exp\left[-\sigma^2(p - k)^2 - \frac{(x - a + pw)^2}{\sigma^2}\right], \tag{82}$$

that is equivalent to $W$ up to a $p-$dependent rescaling of the $x$ variable. In this case expectation values of $p-$dependent operators are invariant under the action of $\hat{U}^-$, i.e., $\langle\psi_-|O(\hat{p})|\psi_-\rangle = \langle\psi|O(\hat{p})|\psi\rangle$, while the expectation value of an $x-$dependent operator $O(\hat{x})$ transforms as

$$\begin{aligned}
\langle\psi_-|O(\hat{x})|\psi_-\rangle &= \int dx \int dp\, O(x)\, W_-(x, p) \\
&= \sqrt{\frac{\sigma^2}{\pi(\sigma^4 + w^2)}} \int dx\, O(x)\exp\left[-\sigma^2\frac{(x - a + wk)^2}{\sigma^4 + w^2}\right] \\
&= \frac{1}{\sqrt{\sigma^2\pi}} \int dy\, O\left(\frac{y - a}{\sigma^2}\sqrt{\sigma^4 + w^2} + a - wk\right) e^{-(y-a)^2/\sigma^2} \\
&= \langle\psi|O\left(\frac{\hat{x} - a}{\sigma^2}\sqrt{\sigma^4 + w^2} + a - wk\right)|\psi\rangle,
\end{aligned} \tag{83}$$

where the second line is found by integrating $W_-(x, p)$ in Eq. (82) with respect to $p$, and the last two lines are obtained by performing the change of variable $x = (y - a)\sigma^{-2}\sqrt{\sigma^4 + w^2} + a - wk$. We deduce that, in case of $x-$dependent operators, the expectation value with respect to $|\psi_-\rangle$ is equivalent to the expectation value with respect to $|\psi\rangle$ where the position operator has been rescaled and shifted according to the final line of Eq. (83). In particular, the first two cumulants of the position operator $\hat{x}$ read

$$\langle x\rangle \equiv \langle\psi_-|\hat{x}|\psi_-\rangle = a - wk,$$

$$\langle x^2\rangle_c \equiv \langle\psi_-|\hat{x}|\psi_-\rangle - \langle\psi_-|\hat{x}^2|\psi_-\rangle^2 = \frac{\sigma^4 + w^2}{2\sigma^2}. \tag{84}$$

The evolution of the state $|\psi_-\rangle$ under the harmonic oscillator dynamics can be explicitly determined from Eq. (25):

$$w(t) = -\frac{\tan(\omega t)}{m\omega},$$

$$\langle x(t)\rangle = a + k\frac{\tan(\omega t)}{m\omega},$$

$$\langle x^2(t)\rangle_c = \frac{(m\omega\sigma^2)^2 + \tan^2(\omega t)}{2(m\omega\sigma)^2}. \tag{85}$$

As a last remark, we note that by combining the results in Eqs. (74), (80), and (85) into the factorised expression for the time evolution of the harmonic oscillator, Eq. (12), we can construct the time evolution of observable, e.g., the position and momentum moments in Eqs. (26).

## C  Time evolution of a Gaussian wave packet

Here we report the detailed calculations of the expectation values of the moments of the position and momentum operators on the Gaussian wave packet in Eq. (16). As a preliminary step to the calculation of Eq. (19), we consider the action of the operator $\hat{U}(t)$ on an eigenstate $|x\rangle$ of the position operator, given by

$$
\begin{aligned}
\hat{U}(t)|x\rangle &= \left\langle \int \frac{dp}{\sqrt{2\pi}}\, e^{\xi^+(t)\hat{x}^2/2}\, e^{i\xi^z(t)\{\hat{x},\hat{p}\}/4}\, e^{\xi^-(t)p^2/2-ipx}|p\rangle \right\rangle_{\phi} \\
&= \left\langle e^{-\xi^z/4} \int \frac{dp}{\sqrt{2\pi}}\, e^{\xi^+(t)\hat{x}^2/2}\, e^{-(\xi^z(t)/2)p\frac{\partial}{\partial p}}\, e^{\xi^-(t)p^2/2-ipx}|p\rangle \right\rangle_{\phi} \\
&= \left\langle e^{-\xi^z/4} \int \frac{dp}{\sqrt{2\pi}}\, e^{\xi^+(t)\hat{x}^2/2} \exp\left[\xi^-(t)\frac{p^2}{2}e^{-\xi^z}-ipxe^{-\xi^z/2}\right]|p\rangle \right\rangle_{\phi} \\
&= \left\langle e^{-\xi^z/4} \int \frac{dp}{\sqrt{2\pi}} \int dy\, e^{\xi^+(t)y^2/2}|y\rangle\langle y|\exp\left[\xi^-(t)\frac{p^2}{2}e^{-\xi^z}-ipxe^{-\xi^z/2}\right]|p\rangle \right\rangle_{\phi} \\
&= \left\langle e^{-\xi^z/4} \int \frac{dy}{\sqrt{2\pi}}\, e^{\xi^+(t)y^2/2} \int \frac{dp}{\sqrt{2\pi}}\exp\left[\xi^-(t)\frac{p^2}{2}e^{-\xi^z}-ipxe^{-\xi^z/2}+ipy\right]|y\rangle \right\rangle_{\phi} \\
&= \left\langle \frac{\exp\left[\xi^z/4+x^2/(2\xi^-)\right]}{\sqrt{-2\pi\xi^-}} \int dy\, \exp\left[\frac{y^2}{2}\left(\frac{e^{\xi^z}}{\xi^-}+\xi^+(t)\right)-y\frac{xe^{\xi^z/2}}{\xi^-}\right]|y\rangle \right\rangle_{\phi},
\end{aligned}
\tag{86}
$$

where $\langle\ldots\rangle_{\phi}$ denotes the expectation value with respect to the Gaussian action $S_0$.

In the first line a completeness relation for the momentum basis, $\int dp\,|p\rangle\langle p| = \mathbb{I}$, was inserted between the last exponential operator and the position eigenket, leading to the appearance of the plane wave $\langle p|x\rangle = e^{-ipx}/\sqrt{2\pi}$, where $\hbar = 1$. In the second line, we substituted $\hat{x}|p\rangle = i\frac{\partial}{\partial p}|p\rangle$ and the consequent action of the dilation operator was written explicitly, i.e., $e^{by\frac{d}{dy}}f(y) = f(e^b y)$. Finally, a further position completeness relation insertion and a Gaussian integration was performed. The convergence of the Gaussian integral is ensured by the fact that the argument of the exponential is purely imaginary. Analogously, the corresponding dual vector evolves according to

$$
\langle x|U^{\dagger}(t) = \left\langle \frac{\exp\left[\bar{\xi}^z/4+x^2/(2\bar{\xi}^-)\right]}{\sqrt{-2\pi\bar{\xi}^-}} \int dz\, \exp\left[\frac{z^2}{2}\left(\frac{e^{\bar{\xi}^z}}{\bar{\xi}^-}+\bar{\xi}^+(t)\right)-z\frac{xe^{\bar{\xi}^z/2}}{\bar{\xi}^-}\right]\langle z| \right\rangle_{\bar{\phi}},
\tag{87}
$$

with $\langle\ldots\rangle_{\bar{\phi}}$ denoting the expectation value with respect to the Gaussian action $S_0[\bar{\phi}]$, and $\bar{\xi} \equiv [\xi(\bar{\phi}(t))]^*$ the complex conjugate of $\xi^{+,-,z}$. Finally, Eq. (19) follows by plugging Eq. (86) into Eq. (16) and integrating the Gaussian integral with respect to $x$.

Finally, the evolution of the wave packet, reported in Eq. (19), is eventually computed by integrating the expression of $\hat{U}(t)|x\rangle$ over the variable $x$ with respect to the Gaussian measure

$$
\exp\left[-(x-a)^2/(2\sigma^2)^2+i(x-a)k\right](\pi\sigma^2)^{-1/4}.
\tag{88}
$$

The convergence is ensured by requiring that $\text{Re}(\gamma) > 0$. In our description we have $\xi^\pm \in i\mathbb{R}$ and real $\xi^z$, so that it is useful to define $\xi^\pm \equiv i\xi_i^\pm$ with $\xi_i^\pm \in \mathbb{R}$, which, together with Eq. (20), leads to

$$\text{Re}(\gamma) = \frac{\sigma^2 e^{\xi^z}}{\sigma^4 + (\xi_i^-)^2} \geq 0. \tag{89}$$

A better understanding of the behavior of $\text{Re}(\gamma)$ can be achieved by considering the following real-valued auxiliary variables:

$$\begin{aligned} X &\equiv \xi_i^- e^{-\xi^z/2}, \\ Y &\equiv e^{-\xi^z/2}. \end{aligned} \tag{90}$$

These variables evolve according to the harmonic equations with time-dependent frequency given in Eq. (6),

$$\begin{aligned} \ddot{X}(t) + \Omega^2(t)X(t) &= 0, \\ \ddot{Y}(t) + \Omega^2(t)Y(t) &= 0, \end{aligned} \tag{91}$$

with initial conditions $X(0) = 0$, $\dot{X}(0) = -m^{-1}$, $Y(0) = 1$ and $\dot{Y}(0) = 0$. These new variables allow one to write $\text{Re}(\gamma) = \sigma^2(\sigma^4 Y^2 + X^2)^{-1}$, making it apparent that $\text{Re}(\gamma(t)) = 0$ if $X(t)$ or $Y(t)$ are infinite. In either case, $\beta(t) = [\sigma^2 Y(t) - iX(t)]^{-1}$ which multiplies Eq. (19), vanishes, i.e., $|\psi(t)\rangle = 0$. Accordingly, the convergence of the Gaussian integral in Eq. (19) is guaranteed by the fact that $\text{Re}(\gamma) \geq 0$ and that whenever $\text{Re}(\gamma) = 0$ the whole $|\psi\rangle$ vanishes.

As a last remark, we point out that the introduction of the variables $X$ and $Y$ in Eq. (90) explains why in the case of the harmonic oscillator, in which the $\xi^{+,-,z}$ are found to be periodically divergent according to the Eqs. (25), there are no divergences in the expectation values of Eqs. (26). In fact, these values depend on a well-behaved combination of the $\xi^{+,-,z}$, satisfying an harmonic equation with constant frequency but different initial conditions.

## D  Derivations of the Dyson Series

In order to prove the equivalence of the Dyson series for the quantum quartic oscillator and the asymptotic expansion of $\hat{U}(t)$ around the harmonic case according to Eq. (32), we begin by calculating the second variation of the time-evolution operator of the harmonic oscillator, which will be useful to determine the first-order correction according to Eq. (32), namely

$$\hat{U}^{(1)}(t) \equiv i\frac{\lambda}{4} \int_0^t ds \frac{\delta^2 \hat{U}_S[\phi]}{\delta\phi(s_1)\delta\phi(s_2)}\bigg|_{\substack{\phi=0 \\ s_1=s_2=s}}. \tag{92}$$

This expression involves the first functional derivative of $\hat{U}_S$, given by

$$\frac{\delta \hat{U}_S[\phi]}{\delta\phi(s)} \equiv G^{(1)}(s|t)\hat{U}_S[\phi], \tag{93}$$

where $G^{(1)}$ is explicitly computed by exploiting the $SU(2)$ commutation relations (10) of the $S$ operators and $\hat{U}_S$, leading to

$$G^{(1)}(s) \equiv \left[\frac{\delta\xi^+}{\delta\phi} - \xi^+ \frac{\delta\xi^z}{\delta\phi} - (\xi^+)^2 e^{-\xi^z}\frac{\delta\xi^-}{\delta\phi}\right]\hat{S}^+ + \left(\frac{\delta\xi^z}{\delta\phi} + 2\xi^+ e^{-\xi^z}\frac{\delta\xi^-}{\delta\phi}\right)\hat{S}^z + \frac{\delta\xi^-}{\delta\phi}e^{-\xi^z}\hat{S}^-, \tag{94}$$

where the parametric dependence on the final time $t$ is understood.

Similarly, the second order functional derivative $\delta^2 \hat{U}_S[\phi]/\delta\phi(s_1)\delta\phi(s_2)$, required to be symmetric under the exchange $s_1 \leftrightarrow s_2$, can be expressed as

$$\frac{\delta^2 \hat{U}_S[\phi]}{\delta\phi(s_1)\delta\phi(s_2)} = \left[G^{(1)}(s_1)G^{(1)}(s_2) + G^{(2)}(s_1, s_2)\right]\hat{U}_S[\phi], \tag{95}$$

where $G^{(2)}(s_1, s_2)$ is found to be

$$G^{(2)} \equiv \left\{\xi^+_{1,2} - \xi^+ \xi^z_{1,2} - \frac{1}{2}\left(\xi^z_1 \xi^+_2 + \xi^z_2 \xi^+_1\right) - \xi^+ e^{-\xi^z}\left[\xi^-_1 \xi^+_2 + \xi^-_2 \xi^+_1 - \frac{\xi^+}{2}\left(\xi^z_1 \xi^-_2 + \xi^z_2 \xi^-_1\right) + \xi^+ \xi^-_{1,2}\right]\right\}\hat{S}^+$$

$$+ e^{-\xi^z}\left[\xi^-_{1,2} - \frac{1}{2}\left(\xi^z_1 \xi^-_2 + \xi^z_2 \xi^-_1\right)\right]\hat{S}^-$$

$$+ \left\{\xi^z_{1,2} + e^{-\xi^z}\left[\xi^-_1 \xi^+_2 + \xi^-_2 \xi^+_1 - \xi^+\left(\xi^z_1 \xi^-_2 + \xi^z_2 \xi^-_1\right) + 2\xi^+ \xi^-_{1,2}\right]\right\}\hat{S}^z; \tag{96}$$

in order to streamline the formulas, the subscripts $\{1, 2\}$ above are used to denote the functional differentiation with respect to $\phi(s_1)$ and $\phi(s_2)$, i.e., $\delta\xi(t)/\delta\phi(s_1) \equiv \xi(s_1|t) = \xi_1$. By taking the functional derivative of Eqs. (13) we obtain a system of differential equations for the first functional derivatives $\xi_{1,2}$, namely

$$i\frac{d}{dt}\xi^+(s|t) + 2\frac{\xi^+}{m}\xi^+(s|t) = 2\delta(t-s),$$

$$i\frac{d}{dt}\xi^z(s|t) + \frac{2}{m}\xi^+(s|t) = 0, \tag{97}$$

$$i\frac{d}{dt}\xi^-(s|t) - \frac{e^{\xi^z}}{m}\xi^z(s|t) = 0,$$

with initial conditions $\xi(s|s) = 0$ for $t \leq s$, reflecting the fact that we assume an Itô-like discretization in deriving Eqs. (13) [20]. The solution to these equations reads

$$\xi^+(s|t) = -\theta(t-s)2i\,\exp\left\{\frac{2i}{m}\int_s^t d\tau\,\xi^+(\tau)\right\},$$

$$\xi^z(s|t) = \theta(t-s)\frac{2i}{m}\int_s^t d\tau\,\xi^+(s|\tau), \tag{98}$$

$$\xi^-(s|t) = -\theta(t-s)\frac{i}{m}\int_s^t d\tau\,\xi^z(s|\tau)\,e^{\xi^z(\tau)}.$$

For $\phi = 0$ they reduce to

$$\left.\frac{\delta\xi^+(t)}{\delta\phi(s)}\right|_{\phi=0} = -i\theta(t-s)2\frac{\cos^2(\omega s)}{\cos^2(\omega t)},$$

$$\left.\frac{\delta\xi^z(t)}{\delta\phi(s)}\right|_{\phi=0} = \theta(t-s)\frac{4}{m\omega}\cos^2(\omega s)\left[\tan(\omega t) - \tan(\omega s)\right], \tag{99}$$

$$\left.\frac{\delta\xi^-(t)}{\delta\phi(s)}\right|_{\phi=0} = -i\theta(t-s)\frac{2}{m^2\omega^2}\cos^2(\omega s)\left[\tan(\omega t) - \tan(\omega s)\right]^2.$$

As expected, the functional derivative $\xi^+(s|t)$ vanishes for $t < s$, as a consequence of the fact that the differential equation at time $t$ does not depend on the realizations of $\phi$ at later times, reflecting the causality of the problem. Following the same line of reasoning as before, the second functional derivatives can be computed directly from their differential equations and

can be expressed in terms of first functional derivative according to

$$\xi_{1,2}^+ = \frac{\xi_1^z \xi_2^+ + \xi_2^z \xi_1^+}{2},$$
$$\xi_{1,2}^z = -e^{-\xi^z(t)}\left(\xi_1^- \xi_2^+ + \xi_2^- \xi_1^+\right), \tag{100}$$
$$\xi_{1,2}^- = \frac{\xi_1^z \xi_2^- + \xi_2^z \xi_1^-}{2},$$

and they are non-zero only if $t > \max(s_1, s_2)$. Moreover, by plugging Eqs. (100) into Eq. (96), we get $G^{(2)}(s_1, s_2) = 0$, such that the only contributing term in the functional derivative in Eq. (95) is

$$G^{(1)}(s)|_{\phi=0} = -i\theta(t-s)x_0^2 \left[\cos(\omega(t-s))\frac{\hat{x}}{x_0} - \sin(\omega(t-s))x_0\hat{p}\right]^2, \tag{101}$$

which finally yields

$$\frac{\delta^2 \hat{U}_S[\phi]}{\delta\phi(s_1)\delta\phi(s_2)}\Bigg|_{\substack{\phi=0,\\s_1=s_2=s}} = -\theta(t-s)x_0^4 \left[\cos(\omega(t-s))\frac{\hat{x}}{x_0} - \sin(\omega(t-s))x_0\hat{p}\right]^4 \hat{U}_0(t)$$
$$= -\theta(t-s)\hat{U}_0(t-s)\,\hat{x}^4\,\hat{U}_0(s), \tag{102}$$

where the time ordering $t > s$ arises naturally from the fact that the equation for $\xi^+$ depends linearly on $\phi$. Collecting the above results of Eqs. (92) and (102), the first-order correction to $\hat{U}(t)$ reads

$$\hat{U}^{(1)} = -i\frac{\lambda}{4}\hat{U}_0(t)\int_0^t \mathrm{d}s\,\hat{U}_0^\dagger(s)\,\hat{x}^4\,\hat{U}_0(s), \tag{103}$$

which is nothing but the first order term in the Dyson series [56]. This can be seen by noticing that the time-evolution operator in the Schrödinger picture $\hat{U}(t)$ can be written in terms of the interaction time-evolution operator in the interaction picture $\hat{U}_I$ as

$$\hat{U}_I(t) \equiv \hat{U}_0^\dagger(t)\hat{U}(t)\hat{U}_0(0), \tag{104}$$

so that, since $\hat{U}_0(0) = \mathbb{I}$,

$$\hat{U}(t) = \hat{U}_0(t)\hat{U}_I(t). \tag{105}$$

The fact that $G^{(2)} = 0$ makes all functional derivatives of order larger than one to depend only on $G^{(1)}$. This allows one to easily generalize the result above to an arbitrary order $n$, leading to

$$\frac{\delta^{2n}\hat{U}_S[\phi]}{\delta\phi(s_1)\cdots\delta\phi(s_{2n})}\Bigg|_{\substack{\phi=0\\s_1=s_2=t_1\\ \cdots \\ s_{2n-1}=s_{2n}=t_n}} = [G^{(1)}(t_1)]^2[G^{(1)}(t_2)]^2\cdots[G^{(1)}(t_n)]^2\hat{U}_0(t), \tag{106}$$

with $t > t_n > t_{n-1} > \cdots > 0$. Using the expression for $G^{(n)}$, this readily yields

$$\hat{U}^{(n)}(t) = \left(-i\frac{\lambda}{4}\right)^n \hat{U}_0(t)\int_0^t \mathrm{d}t_n \int_0^{t_n} \mathrm{d}t_{n-1}\cdots\int_0^{t_2} \mathrm{d}t_1\,\hat{x}^4(t_n)\cdots\hat{x}^4(t_1), \tag{107}$$

where $\hat{x}^4(t) = \hat{U}_0^\dagger(t)\hat{x}^4\hat{U}_0(t)$. Equation (107) is precisely the $n$-th order contribution to the Dyson series in the Schrödinger picture.

# E  Semiclassical limit

In this section we provide details of the computation of the semiclassical approximation for the propagator. The stationary path in Eq. (43) is computed considering the first functional derivative of Eq. (36), namely

$$
\begin{aligned}
\frac{\delta S_{\mathrm{HO}}(y_f, t|y_i, 0)}{\delta\phi(\tau)} &= \frac{\delta}{\delta\phi(\tau)}\frac{m}{2}\int_0^t \mathrm{d}s\left[\dot{y}^2(s) - \Omega^2(s)y^2(s)\right] \\
&= -y^2(\tau) + m\int_0^t \mathrm{d}s\left[\dot{y}(s)\dot{y}_1(\tau|s) - \Omega^2(s)y(s)y_1(\tau|s)\right] \\
&= -y^2(\tau) + m\left[\dot{y}(t)y_1(\tau|t) - \dot{y}(0)y_1(\tau|0)\right] \\
&\qquad - m\int_0^t \mathrm{d}s\, y_1(\tau|s)\left[\ddot{y}(s) + \Omega^2(s)y(s)\right] \\
&= -y^2(\tau),
\end{aligned}
\tag{108}
$$

where $y_1(\tau|s) \equiv \delta y(t)/\delta\phi(\tau)$, $y_f = \sqrt{\lambda}x_f$, $y_i = \sqrt{\lambda}x_i$, and the $t$ dependence is understood. In the third line of Eq. (108) we integrated by parts the right-hand side and finally we exploited Eq. (38) and the explicit expression of $y_1(\tau|s)$

$$
\begin{aligned}
y_1(\tau|s) = &-\left[\frac{\xi_1^-(\tau|t)}{\xi^-(t)} + \frac{\xi_1^z(\tau|s)}{2}\right]y(\tau|t) \\
&+ \frac{e^{-\xi^z(s)/2}}{\xi^-(t)}\left[y_f\, e^{-\xi^z(t)/2}\left(\frac{\xi^-(\tau)}{2}\xi_1^z(\tau|t) + \xi_1^-(\tau|s)\right) + y_i\left(\xi_1^-(\tau|t) - \xi_1^-(\tau|s)\right)\right],
\end{aligned}
\tag{109}
$$

where $\xi_1(t_1|t_2)$ is null if $t_1 \geq t_2$, so that $y_1(\tau|t) = y_1(\tau|0) = 0$. Note that by taking the functional derivative of Eq. (38) computed along the stationary solution $\bar{\phi}$ in Eq. (43) one has

$$
\ddot{\bar{y}}_1(t_2|t_1) + \left[\omega^2 + \frac{\bar{y}^2(t_1)}{m}\right]\bar{y}_1(t_2|t_1) + \frac{2}{m}\bar{y}(t_1)\delta(t_1 - t_2) = 0,
\tag{110}
$$

with boundary conditions $\bar{y}_1(t_2|0) = \dot{\bar{y}}_1(t_2|0) = 0$. The path integral in the second line of Eq. (45) can be expressed in terms of the determinant of the operator $H(t_1, t_2)$, defined in Eq. (46) as

$$
H(t_1, t_2) \equiv \delta(t_1 - t_2) + \frac{1}{2}\frac{\delta^2 S_{\mathrm{HO}}(y_f, t|y_i, 0)}{\delta\phi(t_1)\delta\phi(t_2)}\bigg|_{\bar{\phi}_k} = \delta(t_1 - t_2) - \bar{y}(t_1)\bar{y}_1(t_2|t_1),
\tag{111}
$$

where last equality follows from direct functional derivation of Eq. (108). The determinant of $H$ can be evaluated by relying on the fact that this operator can be recast as the product of two operators whose determinant can be computed exactly. We start by defining the operators

$$
\begin{aligned}
O_1(t_1, t_2) &\equiv \delta(t_1 - t_2)\frac{1}{\bar{y}(t_1)}\left[\frac{d^2}{dt_1^2} + \omega^2 + \frac{\bar{y}^2(t_1)}{m}\right], \\
O_2(t_1, t_2) &\equiv \delta(t_1 - t_2)\frac{1}{\bar{y}(t_1)}\left[\frac{d^2}{dt_1^2} + \omega^2 + 3\frac{\bar{y}^2(t_1)}{m}\right] = O_1(t_1, t_2) + 2\delta(t_1 - t_2)\frac{\bar{y}(t_1)}{m}.
\end{aligned}
\tag{112}
$$

It follows that, given Eqs. (112), one can recast $H(t_1, t_2)$ in Eq. (111) as

$$
H(t_1, t_2) = \int_0^t \mathrm{d}\tau\, O_2(t_1, \tau)O_1^{-1}(\tau, t_2) = \delta(t_1 - t_2) + 2\frac{\bar{y}(t_1)}{m}O_1^{-1}(t_1, t_2),
\tag{113}
$$

where the inverse operator satisfies the following relation

$$\int_0^t d\tau\, O_1(t_1, \tau) O_1^{-1}(\tau, t_2) = \delta(t_1 - t_2).$$
(114)

By plugging Eq. (110) in the inverse operator definition in Eq. (114) we identify $O_1^{-1}(t_1, t_2)$ as

$$O_1^{-1}(t_1, t_2) = -\frac{m}{2} \bar{y}_1(t_2 | t_1).$$
(115)

Hence, we have proved Eq. (111) to be true. It then follows that the path integral evaluates to

$$\int \mathcal{D}\varphi \exp\left\{ -\int_0^t dt_1 \int_0^t dt_2\, \varphi(t_1) H(t_1, t_2) \varphi(t_2) \right\}$$
$$= \left[ \det\left( O_2 O_1^{-1} \right) \right]^{-1/2} = \sqrt{\frac{\det O_1}{\det O_2}} = \sqrt{\frac{f(t)}{F(t)}},$$
(116)

where in the last relation we exploited the fact that $f(t)$ and $F(t)$ are respectively proportional to the determinant of $O_1$ and $O_2$ with the same proportionality constant, according to Eq. (39) and the fact that the determinant of a product of operators is given by the product of the determinants of the individual operators.

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
