# Peer review of "Stochastic Representation of the Quantum Quartic Oscillator"

_SciPost Physics Core, doi:SciPost Phys. Core 6, 029 (2023)_

## Round 1 · Referee Report · Anonymous (Referee 1) · 2023-1-6

Report
This is a very detailed and comprehensive paper (for the brave reader)
on the time evolution of the quartic harmonic oscillator studied in terms of
classical stochastic variables.
A first section introduces the stochastic representation via the introduction
of a ''disentanglement transformation" allowing for the definition of stochastic variables. The dynamics of a Gaussian wave packet is then analyzed and exactly solved limits are considered.
The central part of the paper is probably the stochastic interpretation of section IV and then a perturbative expansion of the time evolution operator is proposed.
According to the authors, the method allow for a ''simple'' and intuitive numerical approach, in spite of possible divergences due to the non-unitarity of the effective evolution operator.
The paper is very well written, with many details of calculations, and very likely useful for readers in the community of open quantum systems. This is nevertheless hard to follow for non experts, but this is due to the technical difficulty of the technique itself.
I recommend publication in the present form. In my opinion, it meets criteria for SciPost Physics but since I am not an expert in the field of open quantum systems, I cannot decide whether it is appropriate for SciPost Physics Core.

---

## Editorial Decision

published